

# Improved Simulation of Thunderstorm Characteristics and Polarimetric Signatures with LIMA 2-Moment Microphysics in AROME

Cloé David[1], Clotilde Augros[1], Benoit Vié[1], François Bouttier[1], and Tony Le Bastard[2]

[1]CNRM, Université de Toulouse, Météo-France, CNRS, Toulouse, France
[2]CNRM, Université de Toulouse, Météo-France, CNRS, Lannion, France

**Correspondence:** Cloé David (cloe.david@meteo.fr)

**Abstract.**

Thunderstorm forecasting remains challenging despite advances in numerical weather prediction (NWP) systems. The microphysics scheme, that represents clouds in the model, partly contributes to the introduction of uncertainties in the simulations. To better understand the discrepancies, synthetic radar data simulated by a radar forward operator (applied to model outputs) are usually compared to dual-polarization radar observations, as they provide insight into the microphysical structure of clouds. However, despite the diversity of microphysics schemes and forward operators, the modelling of polarimetric values and radar signatures such as the $Z_{DR}$ column (ZDRC) remains a complex issue, especially above the freezing level where too low values are often found.

The aim of this work is to assess the ability of the AROME NWP convective model, when coupled with two distinct microphysics schemes (ICE3 1-moment and LIMA partially 2-moment), to accurately reproduce thunderstorms characteristics. A statistical evaluation is conducted on 34 convective days of 2022 using both a global and an object-oriented approach, and a ZDRC detection algorithm is implemented. Simulations performed with LIMA microphysics showed a good agreement with observed $Z_H$, $Z_{DR}$ and $K_{DP}$ below the melting layer in convective cores. Moreover, it demonstrated a remarkable capacity to generate a realistic number of ZDRC, as well as a distribution of (1) the ZDRC area, and (2) the first ZDRC occurrence, very close to the observations. Enhancements in the forward operator have also been suggested to improve the simulations in the mixed phase and cold phase regions.

These findings are highly encouraging in the context of data assimilation, where one could leverage the combination of advanced microphysics schemes and improved forward operators to improve storm forecasts.

## 1 Introduction

Thunderstorms are among the most damaging weather phenomena due to their capacity to generate heavy rainfall, strong winds, hail, and, to a lesser extent, tornadoes. Despite advances in convective numerical weather prediction (NWP) systems, storm forecasting remains challenging. Refining the representation of clouds is a possible area for improvement. In NWP, cloud modelling is done using either a spectral bin or a bulk microphysics scheme. Because of computational costs, single or two-





moment bulk schemes are most commonly used, where hydrometeors are partitioned into distinct categories. Depending on the number of moments, the mixing ratio and/or number concentration evolution is predicted, and physical processes between each category are parametrized. To evaluate microphysics schemes, fine scale observations pertaining to hydrometeors are needed. Since direct hydrometeor observations are relatively rare and mostly confined to field experiments, remote sensing observations, such as radar observations, are generally regarded as a key source of information on hydrometeors as they provide high temporal and spatial resolution data (Houze Jr., 2014). For radar networks that collect dual-polarized data, not only are precipitation intensities accessible via the horizontal reflectivity $Z_H$, but also direct information about hydrometeors such as their axis ratio via differential reflectivity $Z_{DR}$, their liquid water content via the specific differential phase $K_{DP}$ or their heterogeneity inside the radar beam via the cross-correlation coefficient $\rho_{HV}$. Such variables can be used for hydrometeor classification algorithms (Park et al., 2009; Al-Sakka et al., 2013; Besic et al., 2016). Furthermore, these variables offer new insights into thunderstorms, where polarimetric signatures can be identified and attributed to key dynamical and microphysical processes (Höller et al., 1994; Kennedy and Rutledge, 1995; Kennedy et al., 2001; Kumjian and Ryzhkov, 2008).

In recent years, there has been a growing interest in polarimetric data and especially the $Z_{DR}$ column signature, which is defined as a columnar enhancement of differential reflectivity above the environmental freezing level. Primarily explored by Caylor and Illingworth (1987), this signature has attracted the attention of researchers, as it indicates that supercooled raindrops and wet ice particles are lofted by updrafts (Hall et al., 1984; Ryzhkov et al., 1994; Kumjian and Ryzhkov, 2008; Kumjian et al., 2014). In particular, links have been established between $Z_{DR}$ column (ZDRC) and hail growth (Picca and Ryzhkov, 2012; Kumjian et al., 2021). In their study, Kumjian et al. (2021) found that the quantity of hail produced by a storm was directly related to the width of the updraft. They also found that the largest hailstones were generated by stronger and narrower updrafts. The work of Kuster et al. (2019, 2020) has highlighted the usefulness of such a signature in warning decision processes, by analyzing ZDRCs characteristics (depth, area, lifetime) of 42 storms in relation to severe hail and wind reports, including a comparison with currently used reflectivity signatures. Both studies have shown that ZDRCs develop and evolve prior to upper-level $Z_H$ cores, and they provide early signals of changes in updraft strength. Similarly, Lo et al. (2024) studied the ability of ZDRCs to nowcast the evolution of severe convection in the United Kingdom, and came to the same conclusions. A recent study from Aregger et al. (2024) investigated the potential of ZDRCs for hail detection and nowcasting in a complex topography. In particular, the author shows that the maximum $Z_{DR}$ value within the columns was helpful to differentiate the storm types, and that the ZRDC intensifies prior to the first reported hail event. These findings corroborate those of earlier studies, further supporting the potential of ZDRCs to help forecasters issue more accurate severe weather warnings, with improved lead times. In the field of radar data assimilation, Carlin et al. (2017) demonstrated promising outcomes of using ZDRCs to provide moisture and latent heat adjustments, in a modified $Z_H$-based formulation of the Advanced Regional Prediction System cloud analysis (ARPS; Xue et al., 2020, 2001). Likewise, Reimann et al. (2023) indirectly assimilated polarimetric data using microphysical retrievals of the liquid and ice water content and showed improvements on 9-hour precipitation forecasts.

Before attempting to assimilate polarimetric signatures or use them for nowcasting, it is necessary to ensure that models accurately reproduce storm structures and associated polarimetric signatures. Several evaluations have been conducted to assess



model ability to forecast thunderstorms. Typically, such evaluations are performed on fields of precipitation or reflectivities, and
they compare different microphysics schemes against each other and against observations over a few cases (Gallus and Pfeifer,
2008; Rajeevan et al., 2010; Starzec et al., 2018). Now that dual-polarization radar data are becoming more easily available
for the scientific community, evaluations tend to focus on polarimetric variables and signatures. Hence, recent studies have
compared observed to simulated polarimetric signatures with several microphysics schemes, either for an ideal case (Jung et al.,
2010; Johnson et al., 2016) or a few real cases with different storm types (Putnam et al., 2017). Evaluating microphysics allows
a better understanding of the model ability to reproduce known polarimetric signatures and their associated microphysical
processes. For instance, one-moment schemes have been shown to be unable to replicate the $Z_{DR}$ arc in supercells because of
the lack of rime-ice (Johnson et al., 2016) and graupel size sorting (Putnam et al., 2017). Sometimes, comparisons are made
with a single microphysics scheme in order to refine the comprehension of model biases. Thus, sensitivity tests are conducted
as exemplified in the two studies of Shrestha et al. (2022a, b) for a 2-moments scheme including a separate hail class, with
respectively a wintertime stratiform precipitation event and three convective summertime storms. The former study highlighted
that polarimetric signals were underestimated where snow aggregates were dominant in the hydrometeor population between
$-3$ and $-13\,°C$. The latter study showed that the model likewise underestimated the convective area, high reflectivities and the
width/depth of ZRDCs, all of which led to an underestimation of the frequency distribution of high precipitation values.

Going further than the usual point-based forecast evaluation, Davis et al. (2006) was among the first to define an object-
based evaluation framework. The Method for Object-Based Diagnostic Evaluation (MODE) identifies objects by applying
a convolution filter and a threshold on the concerned field. This method has the advantage of not emphasizing a precise
location, thereby facilitating the direct comparison of different features between each other. In Davis et al. (2009) MODE is
applied on precipitation quantities to evaluate 1-h rainfall accumulation from 24-h forecasts, while in Cai and Dumais (2015)
vertically integrated liquid water (VIL) fields are used to evaluate storm characteristics over a 3-week period. This object-
based verification approach can also be leveraged to evaluate the three-dimensional structure of forecasted thunderstorms, as
demonstrated in Starzec et al. (2018). They used observed and forecasted 3D reflectivity fields to define convective objects
where horizontal reflectivities were greater than $45\,\mathrm{dBZ}$. Four months of daily summertime forecasts with either the WRF
single-moment 6-class microphysics scheme (Hong, 2006) or the partially two moment Thompson scheme (Thompson et al.,
2004) have been evaluated. Forecasted objects were found to be more frequent and larger above the melting layer than the
observed objects for both microphysics schemes, but only the two-moment scheme simulated cores with intensities approaching
the observations. One of the first statistical evaluations of polarimetric variables within convective objects was carried out by
Köcher et al. (2022, 2023). The total simulated precipitation, cell core height, and cell maximum reflectivity were analyzed
over a 30-day period. Of the five microphysical schemes evaluated, only one was found to simulate enough convective cells
and none of them were able to reproduce the strongest reflectivities. Three 3-moment bulk schemes showed increased $Z_{DR}$
values above the melting layer, which was attributed to the lofting of large raindrops. This last result suggests that ZDRCs may
have been detected in simulations.

In this context, and preparing for the future use of polarimetric data in the assimilation system of the French operational
convective-scale model AROME (Seity et al., 2011; Brousseau et al., 2016), the present study aims to statistically evaluate





storms structures, not only in terms of accumulated precipitations and polarimetric fields, but also with an object-oriented
approach. Therefore, a ZDRC detection algorithm was developed based on Snyder et al. (2015) and a tracking algorithm
(Heikenfeld et al., 2019) was applied on different convective objects. To the author's knowledge, ZDRC objects characteristics
have never been evaluated this way before. Thus, 34 convective days of 2022 in France were objectively selected from the
European Severe Weather Database (ESWD; Dotzek et al., 2009). The corresponding dual-polarization observations from the
French radar network were compared with synthetic polarimetric data simulated with the Augros et al. (2016) radar forward
operator applied to the AROME model. Simulated variables were obtained from both the operational one-moment microphysics
scheme ICE3 (Pinty and Jabouille, 1998) and the partially two-moment scheme LIMA (Vié et al., 2016), which is intended to
replace ICE3 in a future version of AROME.

Further details regarding the data can be found in section 2. The methodology employed to compare observations and
simulations, as well as a description of the $Z_{DR}$ column algorithm and the tracking algorithm, are explained in section 3.
Results (section 4) are organized as follows : first, a classical model evaluation is performed on accumulated precipitations
fields. Secondly, an object-based evaluation is performed, focusing on convective parts of the storms, where reflectivities, cells
characteristics and polarimetric fields are analyzed. Then, investigations are carried out on $Z_{DR}$ column objects. Finally, results
are discussed in section 5 and the main conclusions of the paper are presented in section 6.

## 2 Data

The following section describes the data used in this study. Precisions about the selected events are given in subsection 2.1
while subsection 2.2 and subsection 2.3 provide more information about the radar and model data compared in this evaluation.
Finally, synthetic radar data are obtained using the forward operator described in subsection 2.4

### 2.1 Studied events

The European Severe Weather Database (ESWD; Dotzek et al., 2009) was used to objectively identify severe convection events.
Reported events undergo quality control procedures concerning the time, location, and the veracity of the report. In this work,
only reports confirmed by reliable sources or scientific case studies have been selected (classified as QC1 or QC2 in the ESWD
database). According to the ESWD, 51 days with hailstones of size equal or greater to 2 cm were reported in France in 2022.
Hence, a total of 34 days have been selected, ranging from April to July, as well as a few days from July to October where
major events occurred in France, such as the Bihucourt tornado on October 23, or the Corsica bow echo on August 18. For each
day, several hail events may occur at different places. In that case, the day is subdivided as necessary by hand. Each selected
period has been cropped to cover the observed event only. In this paper, the term "event" will refer to each individual observed
event (and the associated simulated data). A table listing the studied events, their location, and their duration, can be found in
the appendix Table A1.



## 2.2 Radar observations

Observational data used here comes from the ARAMIS (Application Radar à la Météorologie Infra-Synoptique) operational radar network of Météo-France. In 2025, the metropolitan network consists of 31 dual-polarized Doppler radars (20 C-band, 6 X-band, 5 S-band) covering mainland France. Each dual-polarization radar performs $360°$ scans at 6 elevations angles ranging from a minimum of $0.4°$ to a maximum of $15°$, depending on the radar. There is a supercycle of 15 minutes divided into 3 cycles of 5 minutes each, where the 3 lowest elevations angles are kept the same within the supercycle. The remaining 3

elevations angles vary within each 5 minutes cycle. Raw polarimetric data are corrected through a polarimetric processing chain (Figueras i Ventura et al., 2012). Among the corrections, differential reflectivities are re-calibrated, non-meteorological echoes are identified (Gourley et al., 2007), and attenuation correction is applied to horizontal and differential reflectivities. Hydrometeors are then classified using a fuzzy logic algorithm (Al-Sakka et al., 2013, hereafter A13). The following variables are obtained at a $240\,\mathrm{m}$ x $0.5°$ polar resolution within a maximum radius of $255\,\mathrm{km}$ from each radar, and at a time resolution of

$5\,\mathrm{min}$ : the horizontal reflectivity $Z_H$, the differential reflectivity $Z_{DR}$, the differential phase $\phi_{DP}$, and the co-polar correlation coefficient $\rho_{HV}$. From these variables, the specific differential phase $K_{DP}$ and the echo classification (hereafter ECLASS) are also computed.

In this paper, for the evaluation of precipitation accumulations, ANTILOPE hourly aggregated quantitative precipitation estimations (QPE) are used (Champeaux et al., 2011). ANTILOPE is a composite product where a radar-based precipitation

estimation is merged with rain gauges observations from the Météo-France network. The reader is referred to appendix A of Caumont et al. (2021) for a more detailed description of the ANTILOPE QPE algorithm.

## 2.3 Model data

To compare with observational data, reforecasts of the entire French metropolitan domain have been obtained with the convective-scale NWP model AROME, presented in subsubsection 2.3.1. The two microphysics schemes used in this study are described

in subsubsection 2.3.2 and the radar forward operator used to compute synthetic polarimetric variables from the model output is presented in subsection 2.4. The outputs are generated at the same temporal resolution as the observations (i.e. $5\,\mathrm{min}$).

### 2.3.1 The AROME model

AROME is a non-hydrostatic, deep convection resolving model, running operationally since December 2008 (Seity et al., 2011). This limited-area model is centered on France, and lateral boundary conditions come from the global model ARPEGE

(Action de Recherche Petite Echelle Grande Echelle; Courtier et al., 1991, 1994). In April 2015, the AROME system was upgraded, resulting in (1) an increase in both horizontal and vertical resolutions (from $2.5\,\mathrm{km}$ with 60 pressure levels to $1.3\,\mathrm{km}$ with 90 pressure levels) and (2) a reduction of the data assimilation cycle period from 3 to 1 hour (Brousseau et al., 2016). AROME employs a 3D-Var scheme in a continuous assimilation cycle in order to assimilate mesoscale observations. These include radar radial wind speeds and horizontal reflectivities via a 1D+3D-Var approach (Caumont et al., 2010; Wattrelot

et al., 2014). Since July 2022, 51-hour forecasts are issued at a three-hourly frequency. The AROME model simulates one 2D





prognostic variable (the hydrostatic surface pressure), and twelve 3D prognostic variables: two horizontal wind components, temperature, specific content of water vapor, rain, cloud droplets, snow, graupel, and ice crystals, turbulent kinetic energy, and two nonhydrostatic variables related to pressure and vertical momentum (Bénard et al., 2010). Subgrid processes are parametrized as follows. The surface is modelled with SURFEX (SURFace EXternalisée; Masson et al., 2013) which associates

each grid box of the model with a surface tile (nature, town, sea, or lake), using the ISBA scheme (Interaction Soil Biosphere Atmosphere; Noilhan and Mahfouf, 1996; Noilhan and Planton, 1989) for the natural continental tiles. Local turbulent mixing is computed by a TKE scheme (Turbulent Kinetic Energy; Cuxart et al., 2000) and non-local vertical mixing is performed by a shallow-convection scheme based on a mass-flux scheme (Pergaud et al., 2009). Radiation effects are parametrized by RRTM (Rapid Radiative Transfer Mode; Mlawer et al., 1997) for the long waves and the Fouquart-Morcrette scheme (Fouquart and

Bonnel, 1980; Morcrette and Fouquart, 1986) for the short wave spectrum. The microphysics schemes available are described in the next section. A more detailed documentation of the AROME model is available in Termonia et al. (2018).

### 2.3.2 Microphysics schemes

An accurate representation of thunderstorms requires a mixed-phase microphysics scheme with riming processes and graupel (Seity et al., 2011; Shrestha et al., 2022a). In the AROME model, two bulk microphysics schemes satisfying this requirement

are available : a single-moment scheme, ICE3 (Pinty and Jabouille, 1998), which is used in the operational version of AROME and a flexible two-moments scheme, LIMA (Liquid Ice Multiple Aerosols; Vié et al., 2016; Taufour et al., 2024), currently used for research purposes (Taufour et al., 2018; Ducongé et al., 2020). A description of both schemes is provided below.

The single-moment bulk scheme ICE3 is a three-class ice parameterization coupled to a Kessler scheme (Kessler, 1969) that describes the warm phase. ICE3 manages five prognostic variables of water condensates, in addition to water vapor, to represent

the cloud microphysics. The prognostic equations predict the specific contents ($q$) of three precipitating species, namely rain $q_r$, snow aggregates $q_s$ and graupel $q_g$ (including different types of large-rimed crystals like frozen drops and hail), as well as two non-precipitating species : ice crystals $q_i$ and cloud droplets $q_c$. Furthermore, a generalized gamma distribution is used to represent the particle size distribution (PSD) of each hydrometeor. Power-law relationships are used to link the mass and the terminal speed velocity to the particle diameters. As ICE3 is a one-moment scheme, the total number concentration ($N$)

of each species is diagnostic. For the precipitating species $N_r$, $N_s$, and $N_g$ are deduced from the specific contents, whereas for ice crystals $N_i$ is diagnosed based on the Meyers et al. (1992) parameterization of the heterogeneous nucleation. The total number concentration of cloud droplets $N_c$ is a function of surface characteristics and is set to $300 \ \mathrm{particles/cm^3}$ over land and $100 \ \mathrm{particles/cm^3}$ over sea areas. Finally, ICE3 comes with a subgrid condensation scheme and performs an implicit adjustment of the cloud droplets and ice contents in clouds with a strict saturation criterion. A more complete description with

the associated formulas can be found in Lac et al. (2018).

LIMA is a flexible two-moment bulk scheme with a prognostic representation of aerosols and their interactions with clouds. Aerosol modes are defined by their chemical compositions, PSD, and their capacity to act as cloud condensation nuclei (CCN, parametrised following Cohard et al., 1998), ice freezing nuclei (IFN, parametrised following Phillips et al., 2008, 2013) or coated IFN. LIMA handles the competition between aerosol modes in the activation and nucleation parameterizations.



In our experiments, CCN are initialized with a concentration of $300 \ \mathrm{particles/cm^3}$ and IFN with $1000 \ \mathrm{particles/L}$. The scheme inherits the six water species of ICE3, but in the version of LIMA used here, number concentrations for raindrops $N_r$, ice crystals $N_i$ and cloud droplets $N_c$ are prognostic. The PSD still follows a generalized gamma distribution. As in ICE3, a thermodynamic equilibrium is assumed between the water vapor and cloud droplets. However, the deposition and sublimation rates for ice crystals are computed explicitly based on their mixing ratio and number concentration, allowing

the supersaturation over ice to evolve freely. Some microphysical processes, such as evaporation, melting or homogeneous freezing, have been modified in LIMA to handle the number concentration while others, like the self collection of cloud droplets or the self collection and breakup of raindrops, are entirely new processes. The reader can find a diagram summarizing all the microphysical processes of ICE3 and LIMA in Fig. 7 of Lac et al. (2018). For both schemes, hail can be considered either as a full sixth category or included in the graupel species to form an extended class of heavily rimed ice particles. In this

study, hail is included in the graupel species.

## 2.4  Radar forward operator

Polarimetric variables are not native variables of the AROME model. Consequently, a radar forward operator is required to perform direct comparisons of simulations with dual-polarization observations. The main role of such an operator is to simulate synthetic polarimetric data from model outputs. In this paper, an enhanced version of the Augros et al. (2016) polarimetric

forward operator (hereafter A16) is used. This section will first provide a concise overview of the operator, after which the enhancements will be presented in greater detail.

The AROME outputs, including temperature, pressure and hydrometeor contents, are used as inputs to simulate electromagnetic wave propagation and scattering at all three operational radars wavelengths (S, C and X bands). Back and forward scattering coefficients are calculated for pristine ice particles (considered spherical), rain, snow aggregates and graupel (mod-

elled as oblate spheroids) with the T-matrix method (Mishchenko and Travis, 1994). The scattering coefficients are computed in advance, for different particle sizes, as a function of elevation angle, radar wavelength, temperature, liquid water fraction and hydrometeor type. For each hydrometeor category, the PSD and mass-diameter relationship is provided by the model microphysics. Additionally, the dielectric constant and shape were adjusted for each species, as these parameters are necessary in calculations of the scattering coefficients.

Thanks to the work of Le Bastard (2019), the melting scheme of the A16 radar forward operator was enhanced. In this study, only the graupel can be wet, as hail is included in the graupel class. Wet snow is not simulated, as the microphysics scheme automatically transfers the snow content into the graupel class when it starts melting. To create a (synthetic) wet graupel species, 100 % of the graupel content ($M_g$) provided by the AROME microphysics is converted into the wet graupel content $M_{wg}$, when graupel coexists with rain (i.e. $M_{wg} = M_g$ and $M_g = 0$). There is no limit of temperature, which means that the

melting scheme is also applicable at negative temperature as long as the graupel coexists with rain. This melting scheme is based on the evolution of the liquid water fraction $F_w$, which describes at any time the physical state of the melting particle.





Thus, within the wet graupel, $F_w$ is estimated as a function of the rain and graupel content from the model, $M_r$ and $M_g$, by :

$$F_w = \frac{M_r}{M_g + M_r} \tag{1}$$

The whole melting process is based on a matrix/inclusion approach (see Fig. B1 in Appendix B). At the beginning, the graupel
starts melting, with the melted water first soaking the air cavities (Ryzhkov et al., 2011). Indeed, if the initial density of the dry
graupel (or hail) is less than the density of a whole solid ice sphere of the same diameter, this suggests that the graupel particle
(or hailstone) has air cavities. As stated by Rasmussen and Heymsfield (1987), air cavities are filled from the outside to the
inside of the particle. When all air cavities are filled, a saturated water fraction $F_w^{sat}$ is reached (see Eq. (B13) in Appendix B).
Then, the particle starts forming an outer water shell (Rasmussen and Heymsfield, 1987; Ryzhkov et al., 2011), while the
ongoing mixing of ice and melted water in the core continues to melt. At $F_w = 1$, the graupel is fully melted.

The consequences of the melting process are reflected in both the retrievals of the scattering coefficients, with the computation of an equivalent melt diameter, and the calculation of a new dielectric constant for the wet species. Moreover, a new
particle size distribution is determined following the work of Wolfensberger and Berne (2018). Extended explanations about
the graupel/hail melting process presented here, as well as the formulas of the equivalent melt diameter, the dielectric constant,
and the particle size distribution of the wet specie, are described in Appendix B.

## 3   Methodology

To investigate the impact of the microphysics schemes on the storm structures and the reproduction of polarimetric signatures,
both observational and model datasets have been first pre-processed (subsection 3.1). A $Z_{DR}$ column computation algorithm
has been applied (subsection 3.2) and an object tracking algorithm was used to track storm cells and $Z_{DR}$ columns (subsec-
tion 3.3).

### 3.1   Data pre-processing

As mentioned in subsection 2.2, input radar data contains the $Z_H$, $Z_{DR}$, $\phi_{DP}$, $K_{DP}$, $\rho_{HV}$ and the ECLASS polar fields.
The first pre-processing step consists in removing the gates identified as non-meteorological echoes in the ECLASS field.
Secondarily, a median filter is applied on 3 gates and 3 azimuth angles to reduce the remaining noise in the polarimetric fields.
All the polar radar elevations are then interpolated into a three-dimensional (3D) Cartesian grid with the open-source *Py-ART*
package (Python ARM Radar Toolkit; Helmus and Collis, 2016). The Cartesian 3D grid has a horizontal resolution of $1$ km
and a vertical resolution of $0.5$ km. The grid extends from a height of 0 to $15$ km above ground level and contains all the
aforementioned fields. The *Py-ART* interpolation works as follows : first the radar locations (latitude, longitude, and altitude)
are projected onto the Cartesian grid. Then, for each point of the Cartesian grid, a radius of influence (ROI) is calculated,
determined by the nearest radar. To obtain the value of each Cartesian grid point, all the radar gates values that fall within
the sphere defined by the ROI of the given grid point are summed. Not all radar gates contribute equally : the further the gate
is from the center of the sphere (i.e. the grid point), the less weight its value has when calculating the value of the Cartesian



grid point. *Py-ART* algorithm is flexible, with multiple options of ROI and weighting functions. In this study, to generate a 3D Cartesian grid of polarimetric fields, the Barnes weighting function (Barnes, 1964; Pauley and Wu, 1990) with a ROI based on

virtual beam size options have been preferred. As an additional criterion, a minimum of 3 radars was required to map onto the grid. Indeed, because of the radar scanning strategy (especially for the 3 highest elevations, see subsection 2.2), it is necessary to ensure enough vertical coverage within the studied domain. We have found that combining polar data from at least 3 radars provide enough vertical coverage at high altitudes in the grid.

Forecasts are made on the whole France domain, starting either at 00, 06 or 12 UTC depending on the observed event. To

mitigate the potential spatio-temporal shifts in the model, and because most of this work evaluates storms as objects (so in terms of lifetime and characteristics), the forecasts outputs (1) have been spatially constrained to the observed domain $\pm\, 0.5\,°$ in latitude and longitude, and (2) encompass each observed event to within $\pm\, 2\, \mathrm{hours}$. Simulations outputs are already in a regular horizontal grid configuration. Hence, the horizontal native resolution of $1.3\,\mathrm{km}$ is kept but an interpolation is performed to go from model pressure levels to vertical regularly spaced grid. For easier comparisons with observations, the same $0.5\,\mathrm{km}$

vertical resolution is chosen.

As the last pre-processing step, for both observational and simulation datasets, the freezing level from the forecast is added into the grid. Beforehand, the field is smoothed with a Gaussian filter to avoid local deformations of the $0°\mathrm{C}$ isotherm due to updrafts. It ensures the $Z_{DR}$ column computation is made with the environmental freezing level.

### 3.2 $Z_{DR}$ column detection algorithm

Based on previous work (Snyder et al., 2015; Saunders, 2018; Kuster et al., 2019), a computation of $Z_{DR}$ columns (hereafter ZDRCs) has been implemented and applied on both models and observations pre-processed Cartesian grids. The first step is to apply thresholds on reflectivity ($Z_H$) and differential reflectivity ($Z_{DR}$) 3D fields. To focus on convective cells, $Z_H$ is required to be greater than 25 dBZ, as done in the study of Krause and Klaus (2024). Usually, convective areas are detected for $Z_H > 35$ dBZ (at least). However, the ZDRC is located within the storm's updraft, whereas highest reflectivities are

observed within heavy precipitation regions (storm's downdrafts). As a result, the column appears to be slightly offset from the reflectivity core (see Fig. 8 of Kumjian, 2013) and the $Z_H$ threshold needs to be lowered to ensure the columns won't get cropped. Regarding $Z_{DR}$ values, multiple thresholds have been tested (1, 1.5 and 2 dB) based on the literature. Because of possible remaining biases in the $Z_{DR}$ data and the smoothing induced by the interpolation, a good compromise was found at 2 dB, where no false alarms remain in the observations. Then, all grid points equal to or above the environmental freezing

level are retained, and a 3D boolean mask is created in accordance with all previous requirements. The continuity is verified from 2 km height (not lower because of the curvature of the radar beam) to the top of the column. Nevertheless, holes may exist due to data quality, radar artifacts and/or interpolation method. Thus, a maximum of two missing grid points over the vertical is allowed. Finally, columns that fail to meet the aforementioned criteria are ignored. Relying on Kuster et al. (2019, Fig. 2c), columns whose areal extent were under $4\,\mathrm{km}^2$ or above $150\,\mathrm{km}^2$ have been excluded. Hence, the two-dimensional

ZDRC depth field can be trusted for object tracking.



## 3.3 Object tracking

To track storm cells and $Z_{DR}$ columns, the open-source Python package *tobac* (Tracking and Object-Based Analysis of Clouds; Sokolowsky et al., 2023) has been used. *tobac* consists of a series of functions to apply and customize. A short description of the software is given in subsubsection 3.3.1. Then, subsubsection 3.3.2 details the settings used for the cell tracking, while
settings for the $Z_{DR}$ column tracking are detailed in subsubsection 3.3.3. A summary of all the chosen settings is provided in Appendix C.

### 3.3.1 The tobac software

First, features are identified on a 2D or a 3D field as regions above or below a sequence of thresholds. A feature is represented by its centroid, which is here determined as the center of the region weighted by the distance from the highest detected threshold
value. For example, two thresholds $t_a$ and $t_b$ have been chosen for the feature detection with $t_a < t_b$ . If one $t_b$ contour is detected within the $t_a$ contour, then the centroid is placed inside the $t_b$ contour, otherwise it is placed inside the $t_a$ contour. If two or more $t_b$ contours are identified within the $t_a$ contour (for instance during a storm splitting), then centroids are positioned inside each $t_b$ contour, thereby creating as many objects as $t_b$ contours. For a more visual example, see Fig. 2 of Heikenfeld et al. (2019). As a second optional step, a watershed segmentation can be performed (Carpenter et al., 2006; van der Walt et al.,
2014), based on one given fixed threshold and the previous detected features. This results in a mask with the feature identifier at all pixels identified as part of the object and zeros elsewhere. Again, more details can be found in section 2.3 of Heikenfeld et al. (2019). In this study, only 2D watershed has been performed. The resulting mask can be conveniently used to select the area of each object at a specific time step for further analysis. The last step is the trajectory linking between each feature. Features to link with are looked for in a given search radius. Here the predict method is used (see Fig. 3 of Heikenfeld et al.,
2019) and relies on the previous time step to predict the next position. The reader is referred to Heikenfeld et al. (2019) and Sokolowsky et al. (2023) for a more complete description of the software.

### 3.3.2 Cell tracking

First, tracking of thunderstorms is performed. To detect them in radar observations, two thresholds of 36 and 48 dBZ were chosen and applied on a 2D field of maximum reflectivity over the vertical (hereafter $Z_H^{max\ z}$). The 36 dBZ threshold is helpful
for early detection of the convection, while the 48 dBZ threshold ensure that the convection has installed. As previously stated, the application of two thresholds allows a better storm centroid placement. Consequently, it is necessary to adapt the thresholds to the data type. As shown in the results (in section 4), simulated reflectivities are typically lower. For this reason, the second threshold has been lowered from 48 to 40 dBZ for simulated $Z_H^{max\ z}$. As an additional parameter, a minimum number of contiguous grid points is set, depending on the reflectivity threshold. Thus, for the 36 dBZ threshold, a minimum
of 20 contiguous grid points is required in both observation and simulation datasets, and for the second threshold (40 dBZ in simulations and 48 dBZ in observations), at least 5 grid points are required. If the criterion within the 36 dBZ area is not met, the subsequent 40 − 48 dBZ area cannot be identified. Segmentation is applied twice to obtain, at each time step, a footprint





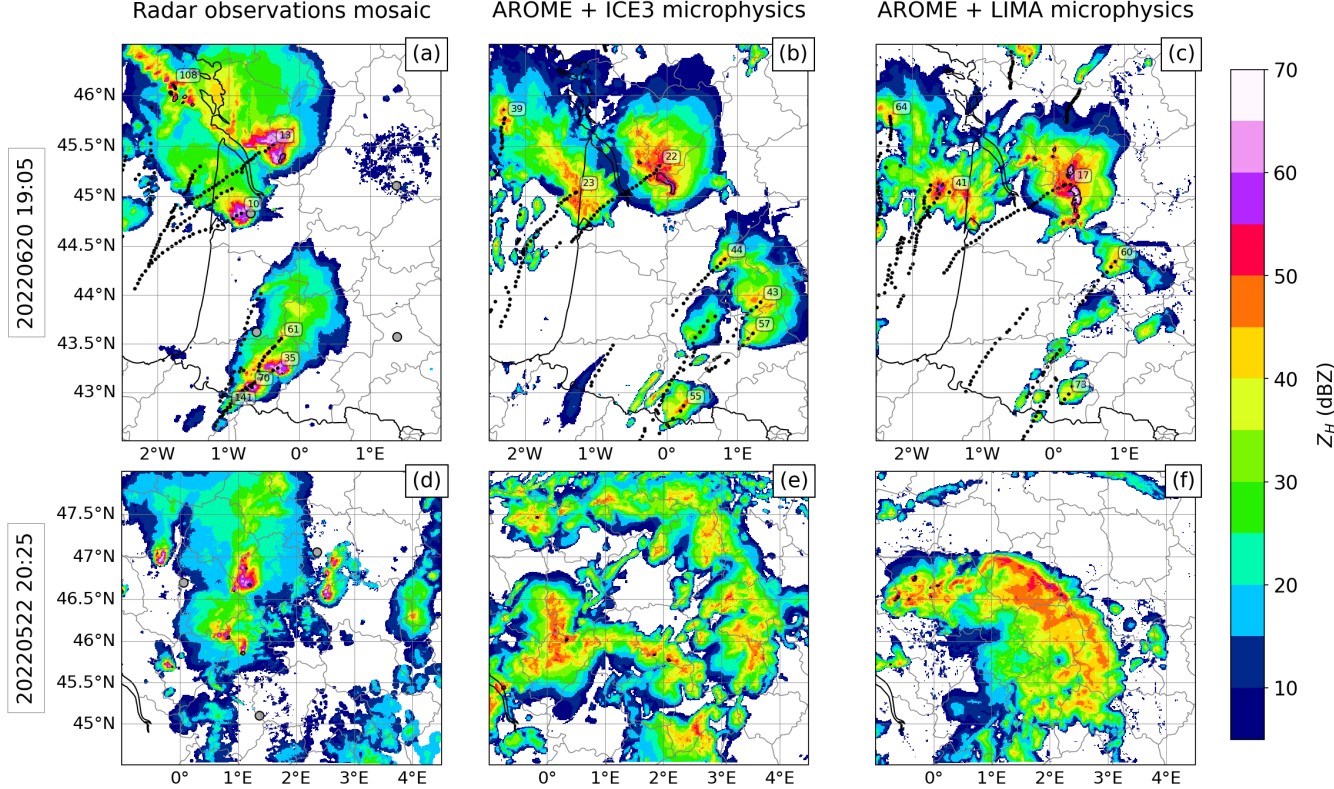

**Figure 1.** Maximum reflectivity over the vertical ($Z_H^{max\ z}$) for two observed **(a, d)** and forecasted convective events, at one time instant, with respectively ICE3 **(b, e)** and LIMA **(c, f)** microphysics schemes. For the first event **(a, b, c)** an example of the tracking algorithm is displayed. Each black dot represent the past positions of the cells centroids. The label associated with the closest dot correspond to the number of the identified cell feature at the time step shown. Each detected $Z_{DR}$ column is materialized with a black contour.

of the cells envelopes and cores. In both observations and simulations datasets, envelopes are defined as contiguous regions of $Z_H^{max\ z} > 25$ dBZ and cores as contiguous regions of $Z_H^{max\ z} > 40$ dBZ. Then the tracking step is carried out to link all the

detected storm cells between each other. To do so, two parameters are adjusted : (1) the radius search range is set to 20 km, and (2) the minimum cell lifetime is set to 9 time steps, which corresponds here to a period of 45 minutes.

    Multiple combinations of thresholds, minimum grid points, and search radius have been tested. The aforementioned parameters proved to be the most relevant for our study. An example of the tracking result is presented in Fig. 1 for the 20th of June 2022. Black accumulated dots represent the identified features (centroids) from the beginning of the event to the showed time

step in the observed $Z_H^{max\ z}$ (Fig. 1a) and in the forecasted $Z_H^{max\ z}$ with either ICE3 (Fig. 1b) or LIMA (Fig. 1c) microphysics.





### 3.3.3 $Z_{DR}$ column tracking

ZDRCs are small and fleeting objects, making them challenging to track. As for the cell tracking, a detection of the columns is applied on the ZDRC field obtained from the algorithm described in subsection 3.2. All ZDRCs greater than $500$ m are detected, and their respective areal extents are determined through the *tobac* segmentation process. Then, the linking of the

ZDRCs centroids is performed with a search radius of $15$ km and a memory of 4 time steps. In the specific case of ZDRCs, the memory parameter has proven to be very useful, as features are allowed to vanish for a certain number of time steps (here 4 time steps i.e. $20$ min) and still get linked into a trajectory by keeping the same object identifier. Finally, to optimize their tracking, ZDRCs have been spatially linked to the identified cells by matching the ZDRC centroid with the cell footprint at each time step.

## 4    Results


This section presents the results obtained from the 34 selected convective days of 2022. A standard model evaluation is conducted, based on accumulated precipitation in subsection 4.1. Then, the evaluation is focused on storm core objects, whose characteristics are investigated in subsection 4.2, and comparisons between observed and simulated polarimetric fields are performed in subsection 4.3. Finally, the evaluation is focused on $Z_{DR}$ column objects in subsection 4.4.

### 4.1    Precipitation


In an effort to objectively evaluate forecasts with respect to observations, skill measures are often computed. Such measures are based on counting observation-forecast "yes-no" pairs to fill in a $2 \times 2$ contingency table that records hits, correct negatives, false positives, and misses. Then, metrics to evaluate the model performance are generated.

     In this section, simulated accumulations of precipitation are compared with ANTILOPE QPE, used as observational refer-

ence (subsection 2.2), on the whole French metropolitan domain and over the same time period. The comparison is performed on a daily basis, over the 34 convective days, and restricted to time periods encompassing precipitating events. The considered events are listed in Table A1. For each day, the aggregation period (for both observations and forecasts) runs from the beginning of the first event to the end of the last event, if multiple events occurred on that day. For instance, two significant events occurred the 3rd of June 2022 : one from 15:00 to 21:00 UTC, in southwest France, and the other one between 16:00 and

22:00 UTC in eastern France. Thus, the total time period considered for the aggregation of cumulative precipitations, for this day, over France (hereafter referred to as RRtot), is from 15:00 to 22:00 UTC (7 hours). For days with only one event listed, the aggregation period corresponds to the duration of the observed event. Four dates are taken into account twice to compute the statistics, in this section only, as the initialization hours of the forecasts are different between the events. These dates are mentioned in bold in Table A1. Indeed, it is not possible to aggregate hourly precipitations issued from different AROME runs.

In this specific case (same day but not same run), the author has ensured that there is no temporal or spatial overlap between the events. Consequently, the number of days considered in this section is artificially increased to 38 (34 + 4).





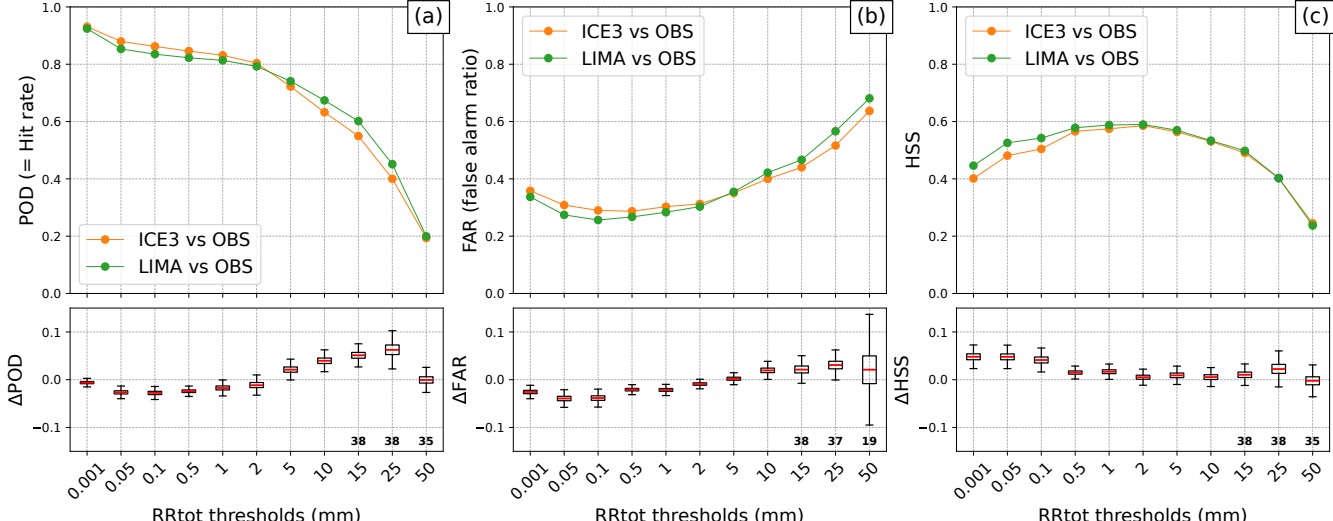

**Figure 2.** POD **(a)**, FAR **(b)** and HSS scores **(c)** for precipitation accumulation forecasts with AROME using ICE3 (orange curves) or LIMA microphysics (green curves). The ANTILOPE QPE is the used as precipitation accumulation reference. For each rain rate threshold and score, the distribution from the bootstrap of the difference between LIMA and ICE3 ($\Delta$) are shown below with box plots. The median is displayed in red. Whiskers indicate the 5th and 95th percentiles of the distribution. For the highest thresholds, the number of days involved in the bootstrap is written in bold.

To fill in the contingency table, the domain has been divided into $50 \times 50$ km boxes. First, the 99th percentile of each box has been calculated for simulated and observed RRtot. This percentile has been chosen (rather than e.g. the median) in order to focus the verification on the convective cells which are usually associated with the heaviest precipitation at each time step. A

set of precipitation thresholds is applied to these RRtot box percentiles, yielding "yes-no" pairs of forecasts and observations that are used to construct the contingency tables (one per accumulation threshold). The tables are summarized by the following scores: the probability of detection (POD; Swets, 1986), the false alarm ratio (FAR; Donaldson et al., 1975), and the Heidke Skill Score (HSS; Heidke, 1926; Murphy and Daan, 1985). These scores are used to compare the forecast performance of the ICE3 one-moment microphysics scheme with the partially two-moment LIMA microphysics scheme (Fig. 2). Additionally, a

bootstrap test is performed to determine if the differences are significant.

The HSS score, which measures the fractional improvement of the forecast over a randomly selected forecast, is presented in Fig. 2c. At the lowest thresholds (RRtot < 2 mm), the HSS is greater for LIMA than ICE3, meaning that LIMA produces better forecasts. This improvement is significant, despite a lower hit rate which is compensated by a lower false alarm ratio (Fig. 2a,b). Regardless of the microphysics, the best forecasts are obtained at intermediate thresholds (RRtot = 1 − 2 mm),

where the HSS is maximum (Fig. 2c). There is no significant difference between both schemes, since the bootstrap confidence intervals are centered close to 0. At the highest thresholds (RRtot > 2 mm), the HSS decreases in both schemes, and LIMA's POD becomes significantly better than ICE3 (Fig. 2a). This can be explained by the fact that LIMA tends to produce higher



accumulations : the total rainfall over the 38 cases reached $1.40 \times 10^5$ m$^3$ for LIMA and $1.29 \times 10^5$ m$^3$ for ICE3, while only

$0.95 \times 10^4$ m$^3$ of rain has been observed. The POD improvement comes at the expense of higher false alarms (Fig. 2b), so that

there is no significant difference in terms of HSS. Hence, the ICE3 and LIMA schemes have similar performance in terms of heavy precipitation forecasts.

## 4.2 Cell characteristics

Although precipitation scores offer some insight into the forecast performance, they can be difficult to interpret in terms of storm behavior. Indeed, this standard approach is influenced by the location and timing of the storms, thereby rendering any

direct comparison of multiple storm objects a challenging undertaking. An object-based approach provides a deeper assessment of cell characteristics, resulting in a more straightforward comparison between observations and simulations, as illustrated below. In particular, comparisons remain valid even when the spatiotemporal structure of the objects are not identical. The object framework is complementary to the standard framework and, together, allows further analysis. Based on the storms objects defined in subsubsection 3.3.2, this section will focus on three key features of a cell: its convective core area, its

duration and its maximum intensity.

To analyze the intensity of storms in terms of objects, the size distribution of all detected convective cores is first studied and presented in Fig. 3. In this figure, the time dimension is ignored, i.e. each core contributes as many times as the number of time steps during which it exists. Figure 3 shows that the small cores (defined here as being smaller than $50$ km$^2$) are too rarely simulated. In the observations, small cores are associated with reflectivity values ranging from $40$ to $68$ dBZ (not shown),

which correspond to emerging/dying cells or weak isolated storms. In the ICE3 and LIMA forecasts, small cores are linked to values of $Z_H \leq 55$ dBZ and $Z_H \leq 64$ dBZ, respectively (not shown). On the other hand, the largest convective cores (defined here as being greater than $1000$ km$^2$) are overestimated in frequency by both forecast models.

As a second key characteristic, the temporal evolution of the thunderstorms is studied. Over the 44 studied events listed in Table A1, a total of 3177 cells were observed and detected, while 1506 cells were simulated with ICE3 microphysics and 1776

with LIMA microphysics. The mean cell lifespan in our observation sample is approximately 75 minutes, while simulated cells last on average 82 minutes with LIMA and 84 minutes with ICE3. Figure 4 shows that there is an underestimation in the number of short-lived cells, defined here as cells that last less than $60$ min from the birth to the death of the convective core. These short-lived cells make up $40$ % of the observational dataset, against $33$ % and $34$ % in the ICE3 and LIMA forecast datasets, respectively. During the mature stage of these observed short-lived cells, at least $65$ % of them had an observed

convective core size that never exceeded $50$ km$^2$, whereas in the forecasts, only $55$ % of the ICE3 and LIMA cells met this condition throughout the mature stage (not shown). The number of convective cells, and in particular the weakest ones, were also underestimated in convective scale simulations with different microphysics schemes in the study of Köcher et al. (2022). These findings demonstrate that neither of the microphysics schemes considered is clearly superior in terms of the size and lifetime of the convective cells, which is consistent with the precipitation scores.

Another key parameter monitored by forecasters is $Z_H^{max\ z}$, the maximum reflectivity within the storm, which is closely related to storm intensity. A study of the $Z_H^{max\ z}$ distribution within each cell core object is herein proposed, with Fig. 5



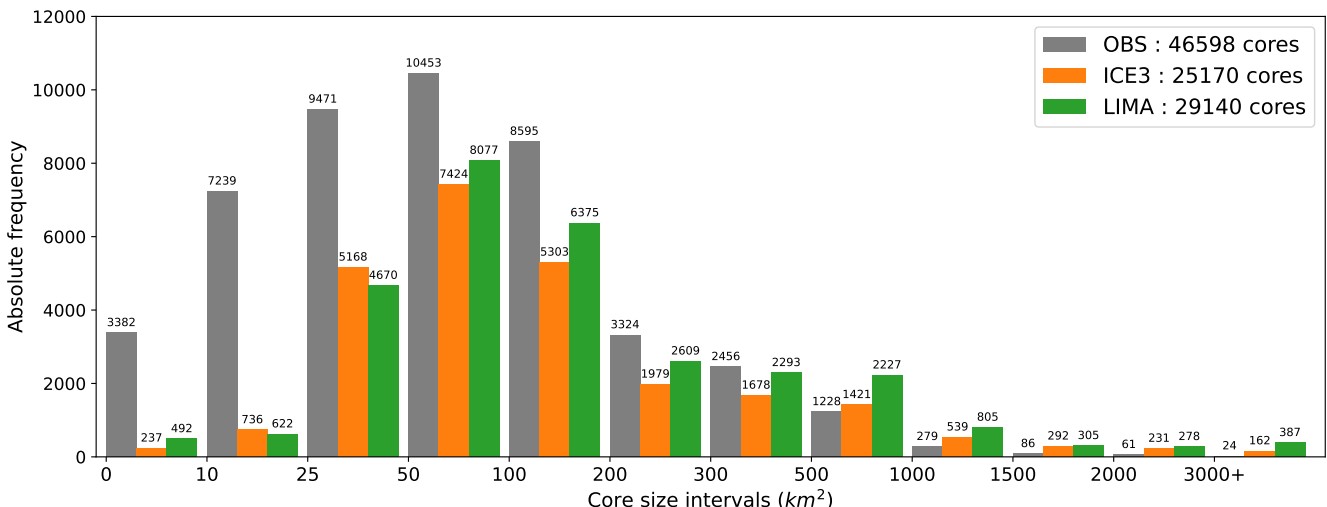

**Figure 3.** Size distribution of detected convective cores ($Z_H^{max\ z} \geq 40\,\mathrm{dBZ}$), over all time steps, for observations (gray bars) and simulations with ICE3 (orange) and LIMA microphysics (green). The total number of detected cores is listed in the legend. The count of each bin is displayed above the bars.

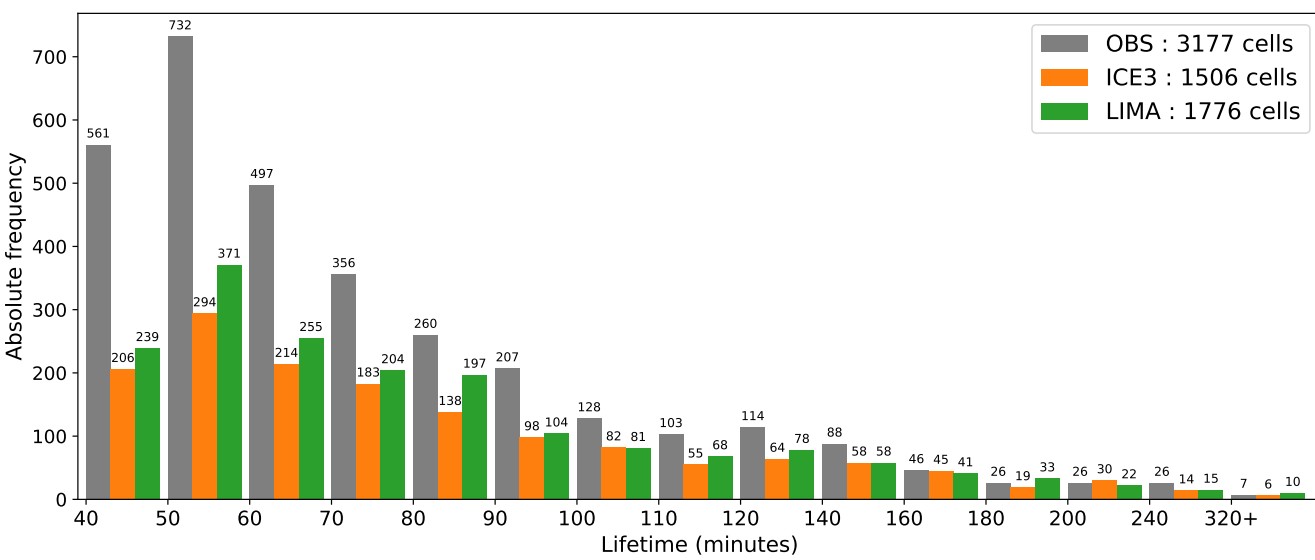

**Figure 4.** Lifetime distribution of detected cells over all convective events, for observations (gray bars) and simulations with ICE3 (orange) and LIMA microphysics (green). The total number of detected cells is listed in the legend. The count of each bin is displayed above the bars.





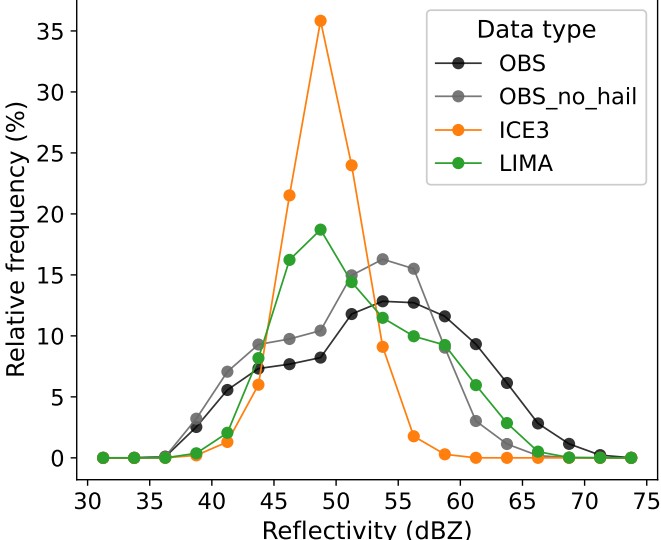

**Figure 5.** Distribution of the cell maximum reflectivity inside the convective core for simulations with ICE3 (orange curve) and LIMA (green curve). The distribution of all observations is displayed in black. The gray curve corresponds to all observations that are not linked with radar-detected hail. Results are given in terms of relative frequency (each point value has been divided by the total number of points, for each data series, and multiplied by 100.)

showing the distribution of the maximum reflectivity a simulated cell can reach compared to its observed counterpart. Since the microphysics schemes considered do not have a distinct hail class, the observation curve (in black, Fig. 5) has been complemented by another curve (in gray) that only includes observed cores not associated with hail as detected by the A13 radar

hydrometeor classification algorithm (see subsection 2.2). Comparing these curves shows that the highest reflectivities are associated with hail detection, in particular the gray curve drops to zero above 65 dBZ, while the black curve does not. The distribution peaks of the ICE3 and LIMA forecasts are similarly located at a reflectivity of 48.5 dBZ, which is lower than in the observations (around 55 dBZ). The $Z_H^{max\ z}$ distribution is narrower in the ICE3 forecasts (orange curve) than in the observations, whereas the LIMA distribution (green curve) is broader. Figure 5 also shows that for cores of $Z_H^{max\ z}$ comprised

between 58 and 68 dBZ, LIMA differs mainly from ICE3 in its ability to simulate high reflectivity. Indeed, over this interval, the LIMA distribution is more consistent with the observations, a conclusion that is even more pronounced when excluding observations associated with radar-detected hail. Cores of the weakest intensities ($Z_H^{max\ z}$ between 35 and 45 dBZ in Fig. 5) are less frequent in the forecasts, regardless of the microphysics involved. This suggests that storms develop and/or die faster in simulations than in observations.





### 4.3 3D analysis of polarimetric fields

This section analyses the 3-dimensional distribution of the reflectivity $Z_H$, the differential reflectivity $Z_{DR}$ and the specific differential phase $K_{DP}$ inside the storm's convective cores (previously defined as $Z_H^{max\ z} \geq 40$ dBZ). Pre-processed polarimetric observations (see subsection 3.1) of these variables are obtained from dual-polarization radars of the Météo-France network. The same variables are simulated by applying the radar forward operator (described in subsection 2.4) on AROME forecasts coupled with either ICE3 or LIMA microphysics. Contoured frequency by altitude diagrams (CFADs; Yuter and Houze, 1995) have been generated for observed and simulated polarimetric fields, inside all detected cores and over all time steps (Fig. 6a, b, and c). In order to gain insight into the processes occurring over the vertical, CFADs are also generated for ICE3 (Fig. 6d) and LIMA (Fig. 6e) hydrometeor contents. The analysis of these results revealed three distinctive regions, which are analyzed separately below.

### 4.3.1 Low-levels

The first region of interest is located under the melting layer, below 3 km height, where most of the observed reflectivities ranged from 40 to 50 dBZ. In this area, $Z_H$ (Fig. 6a), $Z_{DR}$ (Fig. 6b), and $K_{DP}$ (Fig. 6c) decrease with altitude in the LIMA simulations (in green), as observed (in gray), but the distributions are slightly underestimated. In contrast, ICE3 simulations (in orange) show increasing values with height, resulting in strong underestimations at ground level, where the differences between observed and simulated medians are 18 dBZ, 1.2 dB, and 0.5 °/km, respectively. However, in both simulations with ICE3 and LIMA, the rain content increases with height (dark blue curves in Fig. 6d and e) with a steeper slope for ICE3. Figure 6b shows that raindrops are smaller with ICE3, as indicated by the $Z_{DR}$ values which are decreasing towards the ground. Indeed, $Z_{DR}$ is an indicator of the raindrop mean volume diameter (Seliga and Bringi, 1976) and is independent of the drop concentration. Similarly, the $K_{DP}$ values in ICE3 simulations decrease towards the ground (Fig. 6c), as this variable is sensitive to the amount of liquid water. When temperatures are above 0 C°, raindrops evaporate as they fall to the surface. The evaporation process seems to be more efficient in the ICE3 microphysics than in the LIMA microphysics. An explanation for this apparent discrepancy is that LIMA allows the simulation of bigger and more oblate raindrops under the melting layer, because its rain content and number concentration are prognostic. Thus, in similar conditions of humidity, the smaller raindrops in ICE3 will evaporate faster than those in LIMA, leading to a rapid decrease of the rain content in ICE3.

In summary, the rain characteristics are better simulated with the 2-moment representation in LIMA, and vertical profiles of $Z_H$, $Z_{DR}$, and $K_{DP}$ are in stronger agreement with the observations below the melting layer. Moreover, the Q25-Q75 interval shows that the ICE3 $Z_{DR}$ and $K_{DP}$ values have less spread than the observations and the LIMA simulations, the latter having an interval width closer to that observed.

### 4.3.2 Bright-band region

The bright band region (hereafter BB) is a radar signature which highlights the layer where frozen hydrometeors melt to form rain. In our simulations the BB is visible at altitudes between approximately 3 and 3.5 km within convective core objects





(according to the median curves in Fig. 6a, b and c), but not in the observations. Although the BB is not always visible in observed convective areas, other effects such as smoothing due to the beam-broadening (see Meischner et al., 2004, section 2.4.3 and Fig. 10.25 of Ryzhkov and Zrnic (2019)), or the interpolations used to map the radar data onto a 3D Cartesian

grid, can explain the absence of observed BB. The microphysics schemes used in this study do not have prognostic melting species, but a wet graupel species is artificially diagnosed using the enhanced melting scheme implemented in the radar forward operator (subsection 2.4). Figure 6f shows the liquid water fraction $F_w$ within the wet graupel in the detected convective areas. The depth of the layer where its 95th percentile is non-zero shows that wet graupel is present in a broader layer in ICE3 than in LIMA. It can also be seen that the peak of the $F_w$ median reaches higher values in ICE3: at 3 km height, 50 % of the ICE3

wet graupel content is at least 75 % melted, whereas at 3.5 km height, 50 % of the LIMA wet graupel content is at most 15 % melted. These results explain why a more pronounced BB can be seen in ICE3 simulations, especially in terms of $Z_{DR}$ and $K_{DP}$ CFADs (Fig. 6b and c), because these variables are sensitive to the oblateness and liquid water content of the particles.

The differences found between ICE3 and LIMA (in $F_w$ and the BB intensity) may be associated with the broader distribution in both rain and graupel contents, as well as the bigger cloud droplet content in LIMA, in the BB region. However, to better

asses the representation of the BB by both microphysics schemes, an evaluation of stratiform events, where the BB is clearly visible in observations, is required (as in Shrestha et al., 2022a).

### 4.3.3 Above the melting layer

Just above the bright band region (around 4 km), the graupel particles are mostly dry since their liquid water fraction is close to zero (Fig. 6f). Despite significant snow and graupel contents, Fig. 6 shows that simulated $Z_{DR}$ and $K_{DP}$ drop to zero at

levels with low to no rain content, whereas the median of observations remains positive until 8 km for $Z_{DR}$ and 11 km for $K_{DP}$. This is consistent with Köcher et al. (2022) and Shrestha et al. (2022b). A weak positive $Z_{DR}$ value can be associated with dry snow or graupel (which can potentially start to melt), small spherical droplets, and aggregated ice crystals, while a negative $K_{DP}$ may be the sign of vertically oriented ice crystals within a strong electric field (Ryzhkov and Zrnić, 2007; Hubbert et al., 2014). In simulations, only pristine ice, snow and dry graupel are available at higher altitudes (Fig. 6d and e).

However, approximations made in the forward operator results in null values of $Z_{DR}$ and $K_{DP}$ in pristine ice crystals, which are modelled as spheres because of their random orientation (as in Caumont et al., 2006). Very low values of simulated $Z_{DR}$ and $K_{DP}$ could be due to a too low density and/or dielectric constant for dry snow and dry graupel.

On the other hand, observed reflectivities decrease more smoothly with altitude (Fig. 6a). This phenomenon indicates that ice growth processes are underway. Independently of the microphysics, all forecasts showed a sudden decrease of the reflectivity

between 3.5 and 4 km height. Then, at higher altitudes, both microphysics schemes underestimated the reflectivities (LIMA more than ICE3). Compared with the work of Köcher et al. (2022), most of the analyzed microphysics schemes overestimated $Z_H$ values above the melting layer. According to the author, this could be explained either by an overestimation of graupel content or the particle size, or the density set in the forward operator. However, it can be seen with Fig. 6d and e that the main difference between ICE3 and LIMA above the melting layer is the distribution of the ice content, whereas the graupel

distribution is relatively similar. This may be linked with the limitations of the version of LIMA used in this study. Indeed,





**Figure 6.** CFADs of measured and simulated reflectivity **(a)**, differential reflectivity **(b)** and specific differential phase **(c)** inside all detected cell cores ($Z_H^{max\ z} \geq 40$ dBZ). CFADs of the corresponding hydrometeor contents (expressed in $\mathrm{g/m^{-3}}$) for simulations with AROME coupled to ICE3 **(d)** or LIMA **(e)** microphysics scheme. CFAD of the liquid water fraction of the wet graupel within convective cores **(f)** for simulations with ICE3 (orange) or LIMA (green). For all panels, plain lines correspond to the median, and the interval of the 25th and 75th percentiles of the distributions is displayed in lighter colors behind each curve. For panel (f), the 95th percentile is shown with dotted lines. Altitudes are given in km AGL.

ice production is limited by the availability of ice nuclei, and the conversion to snow occurs as soon as pristine ice crystals grow larger than 125 μm in diameter. Thus, ice tends to disappear rapidly, also resulting in lower snow contents than in ICE3. Recent improvements in LIMA (Taufour et al., 2024), such as the implementation of secondary ice production mechanisms, may improve the representation of the upper part of the convective clouds. Although simulations with ICE3 microphysics give

better $Z_H$ values, overall a negative bias still persists when the snow and ice contents are insufficiently represented.



## 4.4 $Z_{DR}$ columns

The use of dual-polarization observations coupled to an object-based detection allows the identification of specific signatures in thunderstorms. This section is focused on the $Z_{DR}$ column (ZDRC), which highlights the location of storms updrafts and indicates the presence of supercooled liquid water being lofted above the environmental freezing level. This signature has raised interest in the scientific community, notably for nowcasting or assimilation purposes (Kuster et al., 2019, 2020; Carlin et al., 2017). Nevertheless, using ZDRCs in such contexts requires them to be correctly simulated by forecast models. This section will concentrate on the main characteristics of a ZDRC : its depth and horizontal extension. As a ZDRC is inherently linked to a convective cell, its temporal evolution will also be examined, as well as its 3D structure.

The first interesting feature about the ZDRC is its maximum height, counted from the $0°C$ isotherm level, and hereafter referred to as column depth. A deep column can be the sign of a very intense updraft, with an increased risk of producing large hailstones, due to a longer residency time above the melting layer, in a zone with enhanced supercooled liquid water content (Kumjian et al., 2021). Figure 7 shows that 69 % of the observed columns had a depth of 0 to 2 km above the $0°C$ isotherm, whereas 31 % of the columns simulated with ICE3 are included in this interval, versus 89 % for LIMA. Columns with a depth greater than 4 km account for 3.1 % of the observed sample and for 2.6 % of the ICE3 sample, which is similar. On the other hand, LIMA only simulated 2 columns with a maximum depth of 4 km. Although the LIMA simulations approximately generated the same total number of columns as the observations (see the legend in Fig. 7), the shape of the observed column depth distribution is not accurately simulated by the two microphysics schemes.

An analysis of the 3D distribution of the differential reflectivity and the hydrometeors contents over the vertical, illustrated by the CFADs within ZDRC objects in Fig. 8, shows one similarity with the results found for the storm core objects : simulated $Z_{DR}$ values (Fig. 8a) rapidly decrease above the freezing level as well as the rain content (dark blue curves in Fig. 8b and c). While dry graupel contents (brown curves) are still significant at this level for both microphysics schemes, the distribution of the liquid water fraction within the wet graupel (Fig. 8d) shows discrepancies between the two schemes. Indeed, within the ZDRCs, rain reached higher altitudes with ICE3 (median at 6 km in Fig. 8b) than with LIMA (median at 5 km in Fig. 8c), although more cloud water (light blue) is available for LIMA (with the 75th percentile going up to 6 km). The distribution of $Z_{DR}$ in Fig. 8a provides additional evidence of the ability of ICE3 to produce deep ZDRCs. However, it can be seen that $Z_{DR}$ values are only slightly larger than 2 dB, the threshold chosen in this study for the ZDRC algorithm. This strict thresholding may penalize the detection of ZDRCs in ICE3 simulations and encourage the selection of only the most intense updrafts, resulting in a low number of detected columns but of a greater depth. In light of the available cloud water in the LIMA simulations, it may be beneficial to consider incorporating this content into the forward operator melting scheme (e.g. taking into account this content in the computation of $F_w$), with the aim of producing slightly higher ZDRCs with LIMA microphysics. Another discrepancy between ICE3 and LIMA is the $Z_{DR}$ signal under 4 km. While $Z_{DR}$ values are underestimated in ICE3 compared to the observation (Fig. 8a), $Z_{DR}$ is overestimated with LIMA whereas the rain content is clearly lower than with ICE3 (Fig. 8b and c). Compared to the observations, such high values of $Z_{DR}$ could be the consequence of a too strong size sorting process within the ZDRCs simulated by LIMA, producing too large (and oblate) raindrops. Similarly, Putnam et al. (2017) and Köcher



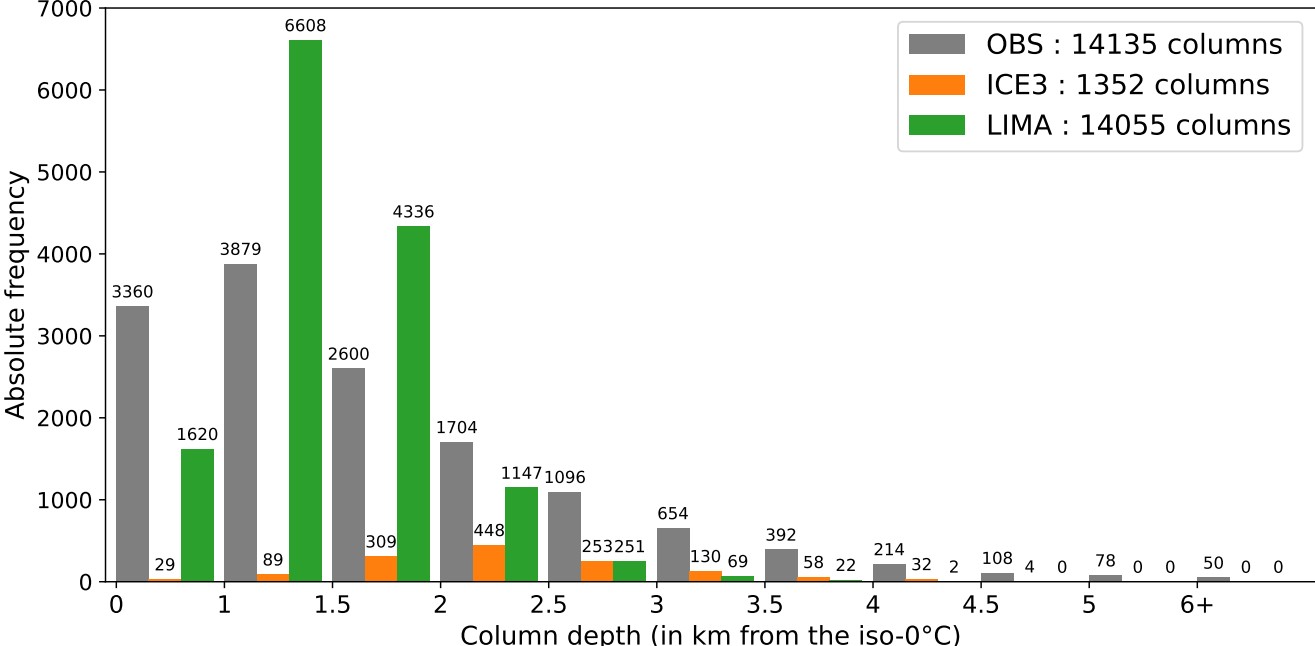

**Figure 7.** Distribution of $Z_{DR}$ columns maximum depth, over all time steps, for observations (gray bars), simulations with ICE3 (orange) and LIMA microphysics (green). The total number of detected columns is listed in the legend. The count of each bin is displayed above the bars.

et al. (2022) found overestimated $Z_{DR}$ values in rain with a 2-moment microphysics, in particular with the Morrison and Thompson schemes using WRF simulations (Skamarock et al., 2008). It should be noted, however, that the overestimation of simulated $Z_{DR}$ in this study is confined to the ZDRC objects and not seen in the convective cores where the $Z_{DR}$ values are very close to the observation and slightly underestimated as discussed in subsection 4.3 and shown in Fig. 6b. Furthermore, the size sorting usually occurs at low levels, whereas the overestimated $Z_{DR}$ values are here mainly located around $3\,\mathrm{km}$ high.

In the microphysics schemes, the size of the raindrops produced by the melting of the graupel is arbitrarily set. Adjusting this parameter could also be a way to reduce the $Z_{DR}$ overestimation just below the melting level.

The study of Kumjian et al. (2021) further highlighted the relevance of the column area as a parameter linked to storm intensification. Indeed, a large column may be indicative of a substantial quantity of supercooled lifted raindrops, which could result in an elevated production of hailstones of small to moderate size. Figure 9 shows the distribution of the column area.

Approximately $64\,\%$ of the observed columns are small features (less than $15\,\mathrm{km}^2$), which is a known characteristic of ZDRC. Simulations with ICE3 demonstrate satisfactory results in terms of proportion, with $69\,\%$ of the ZDRC being small features against $67\,\%$ with LIMA. In Shrestha et al. (2022b), the TSMP model (Gasper et al., 2014; Shrestha and Simmer, 2020) coupled with a 2-moment bulk microphysics scheme (Seifert and Beheng, 2006) was found to underestimate the area of ZDRCs. While in the present study the area of the ZDRCs is not underestimated in terms of proportion, regardless of the microphysics, ICE3



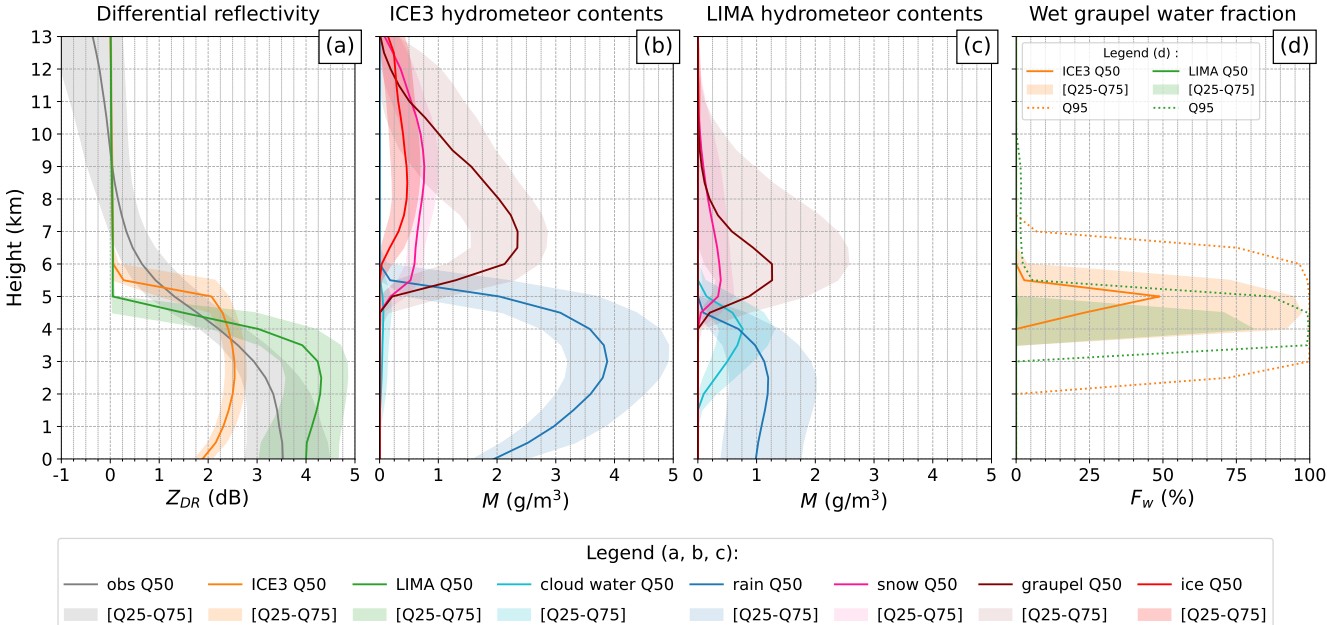

**Figure 8.** CFAD of differential reflectivity (**a**), and CFAD of hydrometeor contents (expressed in $\mathrm{g/m^{-3}}$) in simulations with AROME coupled to ICE3 (**b**) and LIMA microphysics scheme (**c**) inside all detected $Z_{DR}$ columns. Plain lines correspond to the median, and the interval of the 25th and 75th percentiles of the distributions is displayed in lighter colors behind each curve.

simulations nevertheless produced an insufficient number of ZRDCs in comparison to LIMA simulations. The accuracy of LIMA regarding the ZDRC is noteworthy, both in terms of area distribution and absolute number.

As the ZDRC is a proxy of the updraft, it is often observed as soon as the storm core develops. Thus, the first occurrence of a ZDRC is an important marker of the life cycle of the associated storm cell. In this study, $43.8\ \%$ of the observed cells were associated with a ZDRC, against $26.8\ \%$ for ICE3 and $44.5\ \%$ for LIMA. The box plot presented in Fig. 10 shows the

distribution of the first ZDRC occurrence time, relative to the lifetime of the cell to which it belongs. It is expressed as a percentage, where $0\ \%$ correspond to the emergence of the convective core and $100\ \%$ to its death. First, it can be noticed that the 5th and the 25th percentiles are coincident, and equal to $0\ \%$, both in simulations with LIMA and in the observations. This means that in $25\ \%$ of the analyzed events, the ZDRC is formed and detected at the same time as the convective core. The relative lifetime of the cell can be approximately divided as follows: $[0-20\ \%]$ developing stage, $[20-80\ \%]$ mature stage,

$[80-100\ \%]$ dissipating stage. At least half of the first ZDRC detections occur during the early cell development stage. The same behavior is observed in forecasts coupled with LIMA microphysics, but not with ICE3, whose median is around $32\ \%$ of the relative cell lifetime, meaning that more than $50\ \%$ of the columns are detected during the mature and dissipating stage. Although it is uncommon for the initial ZDRC to be observed during the dissipating stage of the cell (gray box plot, Fig. 10), it occurred much more frequently with ICE3, as shown by the broader spread and the absence of outliers in its time distribution

(orange box plot, Fig. 10). Some delay in the ICE3 ZDRC identification may be attributed to the strict thresholding of the



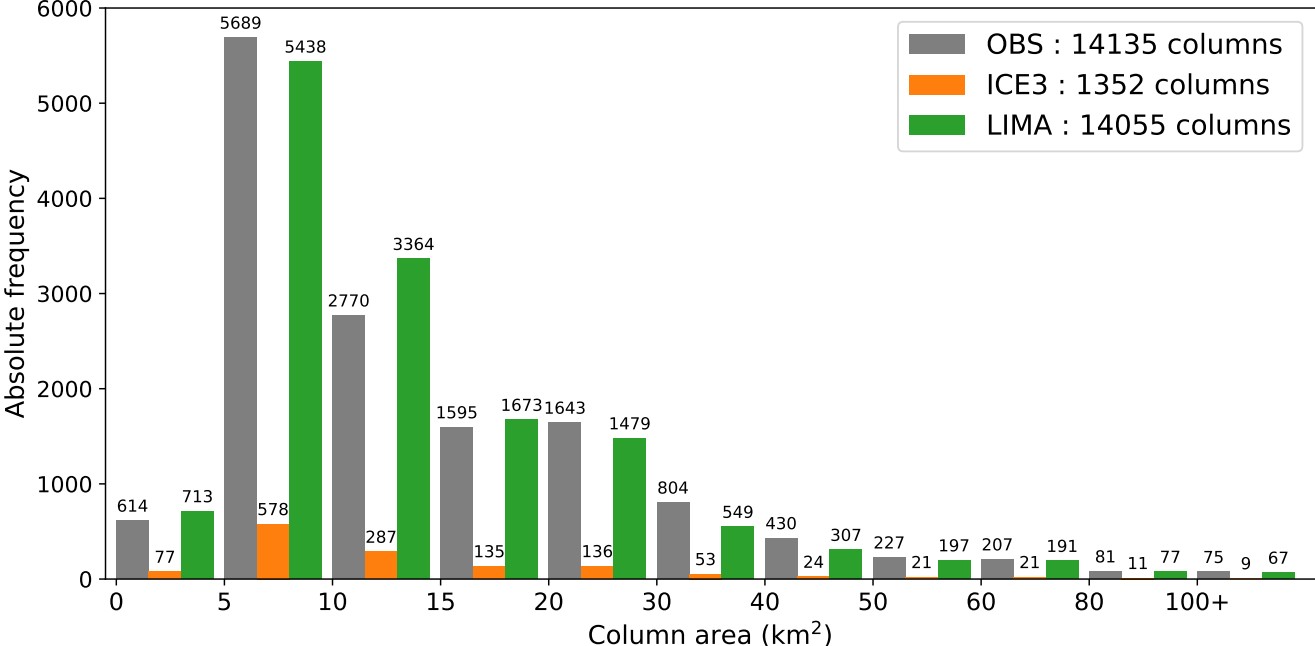

**Figure 9.** Area distribution of $Z_{DR}$ columns, over all time steps, for observations (gray bars), and simulations with ICE3 (orange) and LIMA microphysics (green). The total number of detected columns is listed in the legend. The count of each bin is displayed above the bars. Altitudes are given in km AGL.

detection algorithm (subsection 3.2). One can assume that, in the case where ICE3 detected updrafts are only the strongest, it should counterbalance this effect, since the deepest columns can be expected to occur before the dissipating stage or at the beginning of the mature stage. However, it should be noted that, as seen with Fig. 8b, ICE3 requires very high rain contents to simulate significant $Z_{DR}$ values, because in a 1-moment scheme, the number concentration is unable to vary independently of
the content.

To summarize, the lack of rain above the freezing level within LIMA simulations has a direct impact on the liquid water fraction in the melting scheme (wet graupel), despite the presence of liquid water at these altitudes (cloud water). Moreover, the depth of the ZDRC is not accurately reproduced by neither microphysics scheme, but the ZDRC area, which is a more significant feature, is well reproduced by both schemes in terms of relative distribution. Only LIMA was able to reproduce the
correct amount of ZDRC with a precise area distribution. Finally, the temporal evolution of the ZDRC appears to be accurately forecasted by AROME in conjunction with the LIMA scheme, lending further credibility to this scheme. These findings are highly encouraging with regard to the potential use of LIMA, in particular for the column area feature used with a strict detection threshold.

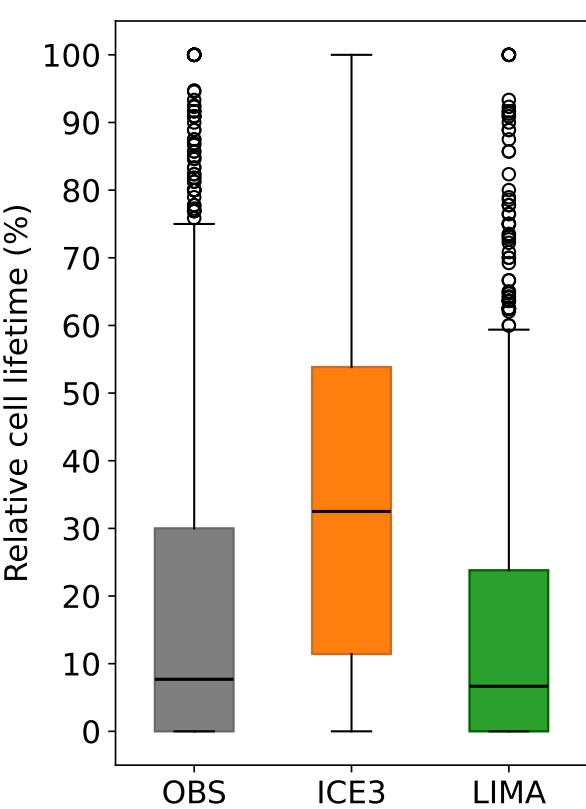

**Figure 10.** First $Z_{DR}$ column occurrence relative to the cell lifetime, for observations (gray bars), and simulations with ICE3 (orange) and LIMA microphysics (green). The cell lifetime is expressed in percentage, 0 % being the emergence of the convective core and 100 % its death. The median is shown with the thick black line. Whiskers corresponds to the 25th and the 75th percentiles. Outliers are displayed as dark circles.



## 5 Discussion

Although the simulations performed with LIMA showed a remarkable agreement with the observations for rain in the convective cores, and a very realistic area distribution and number of ZDRC columns, the present work also revealed discrepancies between observed and synthetic radar data simulated with the A16 forward operator. These discrepancies may be introduced by the radar forward operator, the microphysics schemes, the $Z_{DR}$ column detection method, or a combination of the three. This section is intended as an open discussion about the realistic expectations, the constraints, and possible areas for improvement.

### 5.1 Mixed phase parameterization

The results of this study have emphasized the challenges parameterizing the mixed phase. These issues are characterized by an overestimation of the bright band (BB) in the simulations within convective regions, as well as a notably low amount of rain at negative temperatures, which is particularly pronounced with LIMA microphysics.

To improve the melting layer simulated by the forward operator, it may be beneficial to consider modifying the estimation
of the mixed phase, as various parameterization exist in the literature. For example, Jung et al. (2008) assumed the existence of a mixed phase when dry snow and rain coexist, thus converting a certain proportion of rain and snow to wet snow. On the other hand, in the A16 operator used here, the wet species is computed from the graupel and rain content, when they coexist, and a wet graupel class is created by consuming the entire graupel content only. Wolfensberger and Berne (2018) used an alternative parameterization in which wet aggregates and wet graupel exist within the melting layer. Liu et al. (2024) recently proposed a
melting model, independent of ambient temperature, that relies on the mixing ratio and number concentration of rain and ice species. Such parameterizations will be tested and compared in a future work.

It is worth remembering that, initially, in the A16 forward operator, a wet graupel class was chosen to stay consistent with the microphysics : in the model, the melting snow content is transferred into the graupel class. Because of this, melting graupel output from the forward operator can be either actual melting graupel or melting snow. If the microphysics scheme cannot be
modified, an alternative strategy would be to address the issue directly within the radar forward operator. Therefore, it may be worthwhile to try to differentiate a posteriori melting snow and graupel. In the hydrometeor classification algorithm of Park et al. (2009), a vertical continuity check is performed to distinguish between the convective and stratiform parts of the storm, and has proven to improve the level of discrimination between graupel and snow (dry or wet). Given that the ZDRC detection algorithm developed in this paper includes a vertical continuity check, there is the potential to conduct straightforward tests
to adapt such a continuity check so that a basic discrimination is applied to both the model and observations. Then, this discrimination could be used by the forward operator to separate wet snow and wet graupel.

As a second area for improvement, the computation of the water fraction within wet species could be modified. In the A16 operator, the liquid water fraction is a function of the graupel and rain content, and is equally distributed among the whole particle size distribution. In Dawson et al. (2014) the water fraction is diagnosed with an iterative method. Based on a first
guess, up to 90 % of the rain content available is added and distributed into the melting species contents (graupel and/or hail). A critical mass of water that can exist on a melting particle is estimated, and integrated over the corresponding PSDs, to



determine a critical water fraction which is a function of the particle diameter (according to Fig. 2 in Dawson et al., 2014). If the available rain content exceeds this critical water mass value, then this value is used as the next guess of rain content to be added into the melting species contents. In other words, the liquid water fraction varies across the melting particle size distribution.

In pursuit of greater realism, a two-layer spheroid T-matrix code following Ryzhkov et al. (2011) could also be implemented, as ice, air, and water in a mixed phase particle are currently treated either as a matrix or an inclusion, which is known to affect the computation of the dielectric constant. Nevertheless, Ryzhkov et al. (2013) demonstrated that the T-matrix computation for two-layer and uniformly filled spheroids (assuming water as the matrix) yield similar results in terms of radar variables after an integration over all particle sizes. Moreover, this study was focused on the ability of the model to correctly simulate the radar variables within convective cores, which is not adequate to evaluate the melting scheme parameterization. Thus, improvements in the radar forward operator regarding the bright band signature should be conducted on stratiform events or, at least, focused on improving the stratiform parts of the convective storms. Therefore, the A16 forward operator is currently tested in stratiform environments. While, under the freezing level, realistic polarimetric values were obtained with LIMA, and, with a view to more realistic modelling, it would be valuable to conduct a comparative analysis of the microphysics schemes presented in this work by including a separate hail class, as in this work, hail was included in the graupel class and hence, unable to reach the warmer levels of the atmosphere.

## 5.2 Cold phase modelling

Other challenges have emerged in the cold phase, as simulated $Z_{DR}$ and $K_{DP}$ dropped to almost zero at levels with low to no rain content, despite significant snow and graupel contents. Köcher et al. (2022) and Shrestha et al. (2022b) (hereafter referred to as K22 and S22b) found similar results with different microphysics schemes and radar forward operator parameterizations. Directly above the melting layer, graupel was found to be dominant in both studies. Nevertheless, the aspect ratio of the graupel is assumed to be 1 in K22, while in this study, the aspect ratio is set to 0.8 if the graupel is dry, or is determined using the formulas described in Ryzhkov et al. (2011) if the graupel is wet. In S22b the graupel axis ratio follows the same equations as in this study, but parameterizations of snow and ice are different. Especially, the axis ratio of cloud ice is set to 0.2, against an axis ratio of 1 here. However, even with a 0.2 axis ratio, near-zero $Z_{DR}$ and $K_{DP}$ values have been found within simulations all along the vertical above the melting layer. According to S22b, deficiencies in the forward operator may play a significant role. Shrestha et al. (2022a) showed, for a stratiform event, that snow was found to dominate above the melting layer. In their study, a variety of snow shapes and orientations were tested, but they were still unable to reproduce the observed polarimetric values at these heights. In K22 an enhanced but not realistic $Z_{DR}$ signal has been produced between 8 and 12 km height, with two of the five microphysics schemes analyzed. These high values were caused by the cloud ice, whose assumed aspect ratio by the forward operator is 0.2.

Part of the underestimation of dual-polarization variables in snow could be due to the (too) simple representation of aggregates by homogeneous spheroids with the T-matrix method, as suggested by Trömel et al. (2023), based on the study of Schrom and Kumjian (2018). Another method, known as the discrete dipole approximation (or DDA; Purcell and Pennypacker, 1973;





Draine, 1988), allows the simulation of more complex geometries. This method decomposes the particles into a collection of sub-regions, each being small enough to be considered as a dipole. With this technique, each single dipole can be arranged to form different and complex shapes of crystals or aggregates. However, the DDA method is computationally expensive due to the necessity of solving a large system of linear equations with a dense matrix. It is not implemented in our current forward op-
erator. Hence, in order to increase the $Z_{DR}$ and $K_{DP}$ values in snow, a modification of the snow density-diameter relationship would be more accessible and could be investigated. Moreover, a riming factor could be included in the snow parameterization, following the formulation of Brandes et al. (2007), as it proved to simulate acceptable values in snow in the study of Carlin and Ryzhkov (2019).

Overall, this work, in conjunction with all previous related studies, show how hard it is to simulate accurate polarimetric
values within and above the melting layer, with the diversity of microphysics and forward operator available. Indeed, the ICE3 and LIMA schemes used here simulate very different cloud compositions above the melting layer. ICE3 produced significant amounts of ice crystals, snow and graupel. LIMA, on the other hand, resulted in almost zero ice crystal contents, much less snow and a little less graupel than ICE3. These differences were found both within convective objects and ZDRCs, and suggest that this lies in the treatment of ice particles in the microphysics. In ICE3, ice crystals can be continuously formed in favorable
thermodynamic conditions, whereas in LIMA, ice crystals are formed only when IFN are available. Thus, once IFN are depleted (used to form ice crystals, scavenged by precipitation, etc.), LIMA cannot create new ice crystals. Because of these differences, the pristine ice to snow conversion is based on the ice particle size in LIMA. This process is more efficient than the simple autoconversion process (which is only active for high ice contents) in ICE3. Consequently, pristine ice is quickly consumed in LIMA to form snow that rapidly falls out.

To address this issue, a hybrid configuration of LIMA, using a 1-moment representation of pristine ice crystals as in ICE3, coupled to improvements in the snow representation (Wurtz et al., 2021, 2023), was found to have very good results for tropical convection, retaining most of the advantages resulting from the 2-moment cloud droplets and raindrops. Another solution would be to use a secondary ice production processes (Shrestha et al., 2022a). Such mechanisms (collisional ice break-up and raindrop shattering when freezing) are available in LIMA and have been implemented since the beginning of our study. They
proved very efficient at maintaining significant ice and snow contents in convective clouds. However, as seen within the cold phase, ice crystals are modelled as spheres by the forward operator. To improve the accuracy of the polarimetric outputs and provide a more realistic representation, ice crystals must be modelled as oblate spheroids (Matrosov et al., 1996). More general improvements of the radar forward operator are underway to better represent the ice phase.

### 5.3  $Z_{DR}$ columns

Finally, our ZDRC detection methodology has shown its limitations, especially with regard to the $2\,\mathrm{dB}$ threshold, which contributed to difficulties in the detection of ZDRCs in ICE3. As previously stated, the choice of a high threshold was a commitment made to avoid false alarms in the observations, and hence, to establish a trustworthy reference dataset. Thus, this high $Z_{DR}$ threshold combined with a vertical continuity check partly hindered the detection of ZDRCs in simulations with ICE3 microphysics. Recently, Krause and Klaus (2024) published a novel technique to detect ZDRCs on radar data that only



rely on a single horizontal level (constant altitude plan projection indicators or CAPPI) at the $-10\,°C$ altitude. Their technique consists in identifying the column base rather than the entire column depth. After filtering the data, the median values of two boxes (a 7x7 "outer box" and a 3x3 "inner box") centered on each grid point are compared, yielding the "hotspot" value. Then, any points that exceed the mean hotspot value by more than one standard deviation are retained. Not only did this method demonstrate enhanced performance in regions where there is a bias in differential reflectivities, but this approach is entirely

adaptable to the model world. Thus, the detection of ZDRCs in the model would rely on a comparison of each grid point versus the environment, potentially improving the detection in our simulations with ICE3.

As the last point of this discussion, the detected ZDRCs in the model are essentially composed of wet graupel, thanks to the mixed phase parametrization in our forward operator (i.e. the "wet graupel specie" is artificially created to replace the model "dry graupel" if rain water coexists with graupel, even at negative temperatures). The term "melting scheme" may not

be appropriated to describe the physical state of the graupel at negative temperatures. In fact, in our representation at such temperatures, the graupel is not melting but is probably surrounded by a coat of supercooled liquid water. An alternative method to simulate ZDRCs could be the implementation of a raindrops freezing process in the forward operator, based on Kumjian et al. (2012). As explained in Chapter 7, section 2.5.1 of Ryzhkov and Zrnic (2019), a raindrop freezes from the outside to the inside, and can be represented as an ice shell forming around the droplet, with the core being either purely

liquid or a mix of water and ice (embedded ice germ). Thus, the use of a matrix/inclusion approach to represent this process is absolutely relevant, and could be easily implemented as it is already available in the A16 operator. Nevertheless, given the insufficient amount of supercooled rain water available in our current microphysics schemes, this method may not be able to produce the expected results.

## 6    Conclusion

In this paper, an evaluation of the microphysics of the AROME NWP model was carried out with two different, but complementary, frameworks. Numerical forecasts were compared with observations using a global and an object-based approach. The main objective was to statistically assess the ability of the AROME model to reproduce thunderstorms in terms of intensity, structure, characteristics, and dual-polarization signatures. A related goal was to compare two microphysics schemes against dual-polarization radar observation : the currently operational ICE3 scheme, and the LIMA scheme currently under consider-

ation for future implementation. Given the growing interest of $Z_{DR}$ column in the scientific community due to its potential applications in the fields of nowcasting and data assimilation, this dual-polarization signature was subjected to a careful investigation. To do so, various features were analyzed over a sample of 34 convective days of 2022, and a $Z_{DR}$ column detection and tracking algorithm was implemented. Dual-polarization radar data were extracted from the French metropolitan network. Forecasts were performed with the AROME model coupled to either the ICE3 (1-moment) or LIMA (partially 2-moment)

microphysics schemes. An enhanced version of the radar forward operator of Augros et al. (2016) was applied to the forecasts to obtain synthetic polarimetric fields.





First, a global approach was used to evaluate accumulated precipitation fields. Statistical measures (POD, FAR, and HSS) were computed for different rain rate thresholds and compared between simulations with ICE3 and LIMA. The only significant improvement occurred for the smallest rain rates, with a slight decrease of the POD and the FAR, but a small increase of the HSS score for forecasts with LIMA. For the highest rain rate thresholds, LIMA's overall contribution was fairly neutral as the POD and the FAR both increased, and the HSS remained similar to that of ICE3.

An object-based framework was defined to study polarimetric fields and hydrometeor contents within convective areas of the storms. As a first step, the characteristics of convective cores were compared. Our results shown that the model, regardless of the microphysics, have difficulties simulating small and short-lived cells, which resulted in the generation of larger storms than those observed. Secondly, the 3D structure of polarimetric fields was analyzed with contoured frequency by altitude diagrams (CFADs). The use of LIMA microphysics allowed the simulation of high $Z_H$, $Z_{DR}$ and $K_{DP}$ below the melting layer. This points to a more realistic raindrop sizes representation thanks to the 2-moment component of rain in LIMA. On the other hand, ICE3 (1-moment) raindrops were smaller and, as a consequence, evaporated faster. This caused a reduction of the rain content, which led to a decrease of the $Z_H$, $Z_{DR}$, and $K_{DP}$ variables near the ground. For both microphysics schemes, the bright band was visible, but more pronounced with ICE3. This is mainly due to the creation of a wet graupel species in the radar forward operator, whose liquid water fraction was found to be greater with ICE3 than with LIMA. In the region above the melting layer, both schemes underestimated reflectivities, but ICE3 was more consistent with the observations thanks to a greater amount of snow and pristine ice in the simulations. Nevertheless, simulated $Z_{DR}$ and $K_{DP}$ were found to be null at these altitudes, regardless of the microphysics used. This suggests radar forward operator issues, as (1) pristine ice crystals are currently modelled as spheres, and (2) the snow density and dielectric constant is probably too low, so that the non-spheroidal axis ratio of snow has a minimal influence on the resulting signals.

Finally, the main features of the $Z_{DR}$ column objects were also evaluated. The highest $Z_{DR}$ columns were generated exclusively with ICE3. The analysis of the vertical distribution of $Z_{DR}$ within the columns revealed that ICE3 $Z_{DR}$ values barely exceed 2 dB under the melting layer, but such a value is reached for altitudes up to 5 km versus 4.5 km for LIMA. The lack of raindrops above the melting layer in LIMA's $Z_{DR}$ columns can explain part of this discrepancy. An analysis of the temporal occurrence of the first detected $Z_{DR}$ column, with respect to the cell it belongs to, revealed that most of the columns were detected during the mature and dissipating stage of the cell in ICE3. In contrast, two-thirds of the $Z_{DR}$ columns simulated by LIMA have been detected during the developing stage, as within the observations. Although a strict detection threshold seemed necessary to the author when comparing with observations, as it mitigates the occurrence of false alarms in the observed dataset (so it can be trusted as a reference for the evaluation), the downside is that it only permitted the strongest updrafts in ICE3 simulations to be detected, prompting the selection of regions where a great quantity of rain is lifted above the freezing level. However, this threshold did not hinder the detection of a correct number of columns in LIMA.

As a conclusion, the AROME model, coupled with the partially 2-moment microphysics LIMA, was able to reproduce, on 34 convective days, not only storms characteristics of the convective cells, but also characteristics of much more smaller objects, the $Z_{DR}$ columns. In particular, the simulations with LIMA demonstrated an impressive ability to generate a realistic number of columns, as well as an accurate lifetime and areal extent, which is promising for their future use in both model



evaluation and data assimilation. Previous studies (Li et al., 2017; Carlin et al., 2017; Augros et al., 2018; Putnam et al., 2019; Thomas, 2021; Reimann et al., 2023) have shown possible ways towards assimilation of dual-polarization observations, with different microphysics (1 or 2 moments), different techniques (direct or indirect), and different assimilation schemes (3D-Var,

1D + 3D-Var, EnFK, 1D-EnVar). Notably, these studies have demonstrated the importance of the microphysics scheme and shown limits of the forward operator, mostly in the snow. Likewise, the present study highlighted the need to further improve the Augros et al. (2016) enhanced forward operator and of the two microphysics schemes, especially at and above the melting layer. Thus, in the first place, only the warm phase should be trusted for assimilation purpose, since a lot of work remains on the cold phase, as discussed in this paper. Indeed, to produce more realistic analysis, it is essential to have a model and a

forward operator with the least possible bias. Improving the consistency between observations and the model will contribute to the improvement of background corrections used to compute the analysis.

**Appendix A: List of the studied events**





| Date | Model run | Period (UTC) | Location | Date | Model run | Period (UTC) | Location |
|---|---|---|---|---|---|---|---|
| 2022-04-08 | 12 | 13 to 18 | NE | 2022-06-19 | 12 | 16 to 00 | center-N |
| 2022-04-22 | 06 | 17 to 21 | SW |  |  | 17 to 21 | SW |
| 2022-04-23 | 00 | 08 to 16 | S-SW |  |  | 18 to 22 | SW |
| 2022-05-03 | 06 | 10 to 21 | S | 2022-06-20 | 12 | 17 to 23 | SW |
| 2022-05-04 | 00 | 12 to 19 | E | 2022-06-21 | 00 | 12 to 23 | center |
| 2022-05-15 | 06 | 13 to 22 | SW | **2022-06-22** | 00 | 14 to 01 (+1) | SW |
|  |  | 15 to 00 | N |  | 06 | 09 to 22 | E |
| 2022-05-18 | 06 | 18 to 23 | NW | **2022-06-23** | 06 | 07 to 15 | SW-S |
| **2022-05-20** | 00 | 04 to 11 | NE |  | 12 | 12 to 20 | NE-SE |
|  | 06 | 11 to 19 | NW-N |  |  | 14 to 20 | SW |
| 2022-05-22 | 00 | 16 to 00 | NE |  |  | 16 to 04 (+1) | S-SE |
| 2022-05-23 | 00 | 09 to 13 | E | 2022-06-24 | 00 | 05 to 11 | S |
| 2022-06-01 | 12 | 20 to 10 (+1) | SW | 2022-06-25 | 06 | 13 to 20 | NE |
| 2022-06-02 | 12 | 15 to 00 | center-E | 2022-06-26 | 12 | 13 to 00 | NE |
| 2022-06-03 | 12 | 15 to 21 | SW | 2022-06-28 | 06 | 09 to 19 | S-SW |
|  |  | 16 to 22 | E | 2022-06-30 | 12 | 12 to 19 | NE |
| **2022-06-04** | 06 | 08 to 14 | NW | 2022-07-03 | 00 | 13 to 20 | SE |
|  | 12 | 14 to 22 | center | 2022-07-04 | 12 | 14 to 20 | center-E |
| 2022-06-05 | 00 | 06 to 11 | SE | 2022-07-20 | 12 | 12 to 20 | NE |
| 2022-06-06 | 12 | 14 to 20 | S | 2022-08-16 | 00 | 14 to 23 | S-SE |
| 2022-06-15 | 06 | 15 to 00 | E | 2022-08-18 | 00 | 01 to 08 | SE |
| 2022-06-18 | 12 | 17 to 02 (+1) | SW | 2022-10-23 | 12 | 13 to 18 | NW-N |

**Table A1.** List of the studied events, with the associated date, model run (UTC), period of time, and location for each. These 44 events represent 34 convective days. Dates in bold font indicate that multiple events occurred on the same date, and that the forecasts initializations for each date in bold is different from one event to another. Dates are displayed in YYYY-MM-DD format.





**Appendix B: Description of the enhanced melting scheme of the forward operator**

As previously described in subsection 2.4, non-homogeneous hydrometeors are assimilated to matrices in which inclusions are inserted. The melting process can be divided into five steps (as shown by Fig. B1), based on the estimated liquid water fraction $F_w$ of the particle :

- Initial state : $F_w = 0$, the dry graupel is considered as an ice particle with inclusions of air

- Step 1 : $0 < F_w < F_w^{sat}$, the graupel starts melting, with the melted water first soaking the air cavities from the outside to the inside. This outer layer plays the role of matrix for the whole particle, while the core (still an air matrix with ice inclusions) plays the role of an inclusion. The melting outer layer is considered as a matrix of ice with inclusions of water.

- Step 2 : $F_w = F_w^{sat}$, all air cavities are filled by melted water.

- Step 3 : $F_w^{sat} < F_w < 1$, the particle starts forming an outer water shell, while the core is composed of pure ice and water only. The particle is considered as a matrix of water with inclusions of ice.

- Final state : $F_w = 1$, the whole graupel is melted

Therefore, the estimation of the equivalent diameter $D_{eq}$ that is needed to retrieve the scattering coefficients, as well as the dielectric constant, are tailored to the two stages of the melting process.

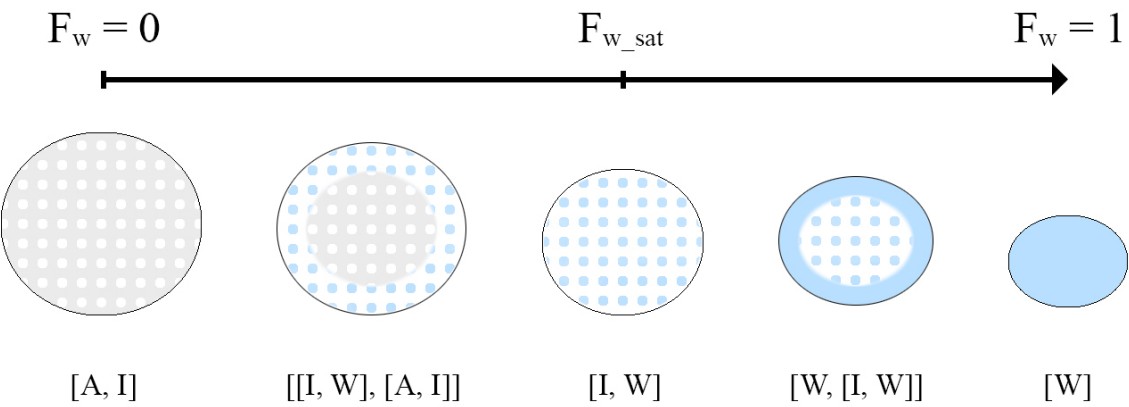

**Figure B1.** Illustration of the graupel melting process as a function of the liquid water fraction $F_w$, with a matrix/inclusion approach. Air (A), ice (I) and liquid water (W) are shown respectively in gray, white, and blue. The various combinations [matrix, inclusions] used to calculate the dielectric constant according to the Maxwell-Garnett approximation (see 2.4) are also shown. Taken with permission from Fig. 4.1 of Le Bastard (2019).





## B1  Equivalent diameter of melting graupel

We want to remind the reader that the only information accessible by the forward operator are the variables related to the dry

species. Thus, $\rho_g^{(i)}$ will refer to the initial graupel density, which is the density of the dry graupel provided by the model. $\rho_g^{(i)}$ includes implicitly the contribution of air, which means that $\rho_g^{(i)} < \rho_i$ (with $\rho_i$ being the density of pure ice) if there are air cavities inside the particle.

### B1.1   $0 \leq F_w < F_w^{sat}$

To compute the scattering coefficients of the melting graupel, a spherical melted equivalent diameter is first estimated. The wet

graupel volume $V_{wg}$ is equal to the volume of a sphere of equivalent diameter $D_{eq}$ :

$$V_{wg} = \frac{\pi}{6} D_{eq}^3 \tag{B1}$$

As the graupel melts, $V_{wg}$ can be expressed as a function of the initial volume of the dry graupel $V_g^{(i)}$ (of equivalent spherical diameter $D$) and the volume of melted graupel $V_{mg}$ :

$$V_{wg} = V_g^{(i)} - V_{mg} \text{ , with } V_g^{(i)} = \frac{\pi}{6} D^3 \tag{B2}$$

By conservation of mass, the mass of melted graupel $m_{mg}$ is equal to the mass of water created $m_w$ :

$$m_{mg} = m_w \Leftrightarrow V_{mg} \times \rho_g = m_w \tag{B3}$$

Also, the water fraction can be defined as the mass of water out of the total mass of the wet graupel. In this operator, $m_{wg} = m_g^{(i)}$, thus, the mass of water created is :

$$m_w = F_w \times m_g^{(i)} \tag{B4}$$

By combining equations B1, B2, B3, and B4, the equivalent diameter $D_{eq}$ for $F_w \in [0, F_w^{sat}]$ can finally be expressed as :

$$D_{eq} = D \times (1 - F_w)^{1/3} \tag{B5}$$

### B1.2   Estimation of $F_w^{sat}$

The saturation is reached when the air cavities are fully soaked by water. The volume of the cavities $V_c$ depends on the liquid water fraction $F_w$. In other words, $F_w = F_w^{sat}$ when $V_c = 0$.

First, we need to determine the initial volume of the cavities $V_c^{(i)}$, when the particle is dry (at $F_w = 0$) i.e. only made of air and ice.

$$\begin{cases} V_c^{(i)} = V_g^{(i)} - V_i \\ m_g^{(i)} = m_i \end{cases} \Rightarrow V_c^{(i)} = V_g^{(i)} \left( 1 - \frac{\rho_g^{(i)}}{\rho_i} \right) \tag{B6}$$





Then, when the graupel starts melting, the new volume of air cavities $V_c^{(*)}$ can be expressed as

$$\forall F_w \in ]0; F_w^{sat}], \ V_c^{(*)} = V_{wg} - V_i - V_w \tag{B7}$$

$V_i$ can be deduced by the conservation of the mass :

$$m_{wg} = m_g^{(i)} = m_i + m_w \Rightarrow V_i = V_g^{(i)} \frac{\rho_g^{(i)}}{\rho_i} - V_w \frac{\rho_w}{\rho_i} \tag{B8}$$

The expression of $V_w$ comes directly from Eq. (B4) :

$$V_w = F_w V_g^{(i)} \frac{\rho_g^{(i)}}{\rho_w} \tag{B9}$$

By replacing $V_i$ and $V_w$ in Eq. (B7) :

$$V_c^{(*)} = V_g^{(i)} \left(1 - \frac{\rho_g^{(i)}}{\rho_i}\right) + \frac{\rho_g^{(i)}}{\rho_i} F_w V_g^{(i)} - \frac{\rho_g^{(i)}}{\rho_w} F_w V_g^{(i)} \tag{B10}$$

Finally, the volume of the air cavities for a liquid water fraction comprised between $0$ and $F_w^{sat}$, is the initially available volume from which the new volume occupied is subtracted.

$$V_c = V_c^{(i)} - V_c^{(*)} \tag{B11}$$

$$= V_g^{(i)}(1 - F_w) \left(1 - \frac{\rho_g^{(i)}}{\rho_i}\right) - V_g^{(i)} F_w \frac{\rho_g^{(i)}}{\rho_w} \tag{B12}$$

As previously stated, saturation ($F_w = F_w^{sat}$) is reached for $V_c = 0$. Thus, with Eq. (B12) it finally comes that :

$$F_w^{sat} = \frac{\frac{1}{\rho_i} - \frac{1}{\rho_g^{(i)}}}{\frac{1}{\rho_i} - \frac{1}{\rho_g^{(i)}} - \frac{1}{\rho_w}} \tag{B13}$$

**B1.3  $F_w^{sat} \leq F_w \leq 1$**

Once fully soaked, the particle starts building a water shell. At this stage, the graupel is composed of pure ice and water only. The volume of the wet graupel is thus :

$$V_{wg} = V_i + V_w \tag{B14}$$

Its mass $m_{wg}$ can be expressed as a function of the mass of ice and the mass of water, including the water within the core and in the shell, by conservation of mass :

$$m_{wg} = m_g = m_i + m_w \Rightarrow m_g = m_i + F_w m_g \text{ using Eq. (B4)} \tag{B15}$$

By expressing $m_i$ as a function of $m_g$, Eq. (B14) can be written :

$$V_{wg} = \frac{\pi}{6} D^3 \rho_g \left(\frac{1 - F_w}{\rho_i} + \frac{F_w}{\rho_w}\right) \tag{B16}$$

Finally, with Eq. (B1) and Eq. (B16), the equivalent diameter for $F_w > F_w^{sat}$ is :

$$D_{eq} = D \times \left[\rho_g \left(\frac{1 - F_w}{\rho_i} + \frac{F_w}{\rho_w}\right)\right]^{1/3} \tag{B17}$$



## B2 Dielectric constant computation

The Maxwell Garnett (1904) formulation can be used to define the permittivity of the mixed phased particles, as long as
the inclusions are spherical and remain small compared with the wavelength. Let $\epsilon_{mat}$ be the matrix permittivity and $\epsilon_{inc}$
the permittivity of inclusions. Following Maxwell Garnett (1904) and Wolfensberger and Berne (2018), the permittivity of
non-homogeneous hydrometeors can be expressed as :

$$\epsilon(\epsilon_{mat}, \epsilon_{inc}, f) = \epsilon_{mat} \left( \frac{1 + 2f\alpha}{1 - f\alpha} \right) \text{, with } \alpha = \frac{\epsilon_{inc} - \epsilon_{mat}}{\epsilon_{inc} + 2\epsilon_{mat}} \text{ and } f = 1 - \frac{\rho}{\rho_{mat}} \text{ the inclusions volume fraction} \tag{B18}$$

Based on Fig. B1, the effective permittivity can be expressed as :

$$\epsilon_{wg} = \begin{cases} \epsilon(\epsilon(\epsilon_i, \epsilon_w, f_{iw}), \epsilon(\epsilon_a, \epsilon_i, f_{ai}), f_{iw/ai}) & \text{if } F_w \leq F_w^{sat} \\ \epsilon(\epsilon_w, \epsilon(\epsilon_i, \epsilon_w, f_{iw}), f_{w/iw}) & \text{if } F_w > F_w^{sat} \end{cases} \tag{B19}$$

With the volume fractions involved in the calculation being respectively :

$$\begin{cases} f_{iw} = 1 - \dfrac{\rho_g}{\rho_i} \\ f_{ai} = \dfrac{\rho_g}{\rho_i} \\ f_{iw/ai} = 1 - \dfrac{F_w}{1 - F_w} \dfrac{\rho_i \rho_g}{\rho_i \rho_w - \rho_g \rho_w} \\ f_{w/iw} = \dfrac{(1 - F_w)}{\rho_g \left( \frac{1 - F_w}{\rho_i} + \frac{F_w}{\rho_w} \right)} \end{cases} \tag{B20}$$

## B3 PSD for melting hydrometeors

As the liquid water fraction increases, the particle size distribution gradually changes from that of iced particles to that of rain.
The flux-based approach used here comes from Szyrmer and Zawadzki (1999). Given the hypothesis of no aggregation of the
snowflakes and no breakup of the raindrops, at a stationary state, one iced particle leads to one raindrop as the particle melts.
Thus, it can be stated that the flow of particles is conserved (subscripts $_{wp}$ and $_p$ stand for "wet particle" and "dry particle",
respectively) :

$$N_r(D_{eq}) v_r(D_{eq}) = N_{wp}(D_{eq}) v_{wp}(D_{eq}) = N_p(D_{eq}) v_p(D_{eq}) \tag{B21}$$

with $N$ being the number concentration and $v$ the terminal velocity of the corresponding hydrometeors. The terminal veloc-
ities of wet particles are computed from a combination of the terminal velocities of rain $v_r$ and dry particles $v_p$, following
Wolfensberger and Berne (2018) :

$$v_{wp} = \Phi v_r + (1 - \Phi) v_p \text{ , with } \Phi = 0.246 F_w + (1 - 0.246)(F_w)^7 \tag{B22}$$

Then, knowing $F_w$ for each $D_{eq}$, the total wet particle number concentration $N_{wp}$ can be expressed :

$$N_{wp}(D_{eq}) = (1 - F_w) \frac{v_p}{v_{wp}} N_p(D_{eq}) + F_w \frac{v_r}{v_{wp}} N_r(D_{eq}) \tag{B23}$$



## Appendix C: Summary of *tobac* settings

|  | cell envelope | cell core | | $Z_{DR}$ column |
|---|---|---|---|---|
| **Features detection** |  | *obs* | *model* |  |
| Threshold(s) | / | 36 and 48 dBZ | 36 and 40 dBZ | 500 m |
| Minimum object(s) size | / | 20 and 5 km$^2$ | 23 and 6.5 km$^2$ | 4 km$^2$ |
| **Segmentation** |  |  |  |  |
| Contour threshold | 25 dBZ | 40 dBZ | | 2 dB |
| **Tracking** |  |  |  |  |
| Search radius | / | 20 km | | 15 km |
| Memory | / | 5 min | | 20 min |
| Minimum lifetime | / | 45 min | | 5 min |

**Table C1.** Settings of the open-source Python package *tobac*, used in the tracking step, for cell envelope, cell core and $Z_{DR}$ columns objects

*Author contributions.* CD designed the methodology, developed the $Z_{DR}$ column algorithm, realized the analysis, and wrote the manuscript. CA, BV and FB helped in the conceptualisation, methodology, and the supervision of the manuscript. TLB implemented the operator's melting scheme and contributed to its description in Appendix B

840   *Competing interests.* The authors declare that they have no conflict of interest.

*Acknowledgements.* This study has been partially supported through the grant EUR TESS N°ANR-18-EURE-0018 in the framework of the Programme des Investissements d'Avenir. The authors would like to thank Pr. Alexander Ryzhkov, Dr Jacob Carlin, and Dr Jeffrey Snyder for the discussions and constructive feedback that helped us better understand our limitations and pave the way for future prospects. The authors would also like to express their gratitude to John Krause for his review of the manuscript, which enhanced its overall quality.



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
