# Peer review of "Improved Simulation of Thunderstorm Characteristics and Polarimetric Signatures with LIMA 2-Moment Microphysics in AROME"

_EGUsphere, 2025_

## Author Comment (AC2)

**Review of egusphere-2025-685 #1**

**Indications of authors :** The authors would like to express their gratitude to the anonymous referee for his/her meticulous examination of the paper, which will undoubtedly enhance its quality. We hope that readers will find the answers useful.

The response to each comment is indicated in blue. Citations to parts of the paper will be quoted, and, if necessary, modified text according to the reviewer's comments will be underlined.
* * *
**General comments:**

The paper analyzes the ability of the AROME NVP convection model to represent convective thunderstorms. The model is run with two different microphysical schemes, which are validated on the basis of the French polarimetric radar network for 34 convective days in 2022.

In general, the paper is well written, the work is comprehensive but also quite lengthy as a result. However, I would not have a direct suggestion on how it should be shortened: The introduction is well researched, chapters 2 and 3 are mandatory for data and methodology, and the results section with comparisons of a) precipitation b) polarimetric moments c) ZDR columns is adequate. I also liked the appendix with the more technical insights into the forward operators melting scheme. Nevertheless, I would have a few minor points regarding clarity. Otherwise, I am pleasantly surprised that I found hardly any technical corrections.
* * *
**Specific comments:**

**L117:** Why were not all 51 days selected for this study? Is it because only 34 days of these 51 days have QC1 or QC2? That is not entirely clear here.

We understand that this is confusing. In fact, it took a lot of time to get from the archives the radar data, and to replay them. Furthermore, we also had to reforecast each studied day two times (one per microphysics). Since we worked with a temporal resolution of five minutes (so our model data was consistent with our radar data), it generated a huge amount of data to analyze. Therefore, we decided to focus on events that occurred during spring and early summer (i.e. from April to July). However, 2022 had been a year rich in strong convection events, and because we were aiming to have a wide diversity of study cases, we finally decided to include 3 more study cases in our analysis sample : a major thunderstorm event all across France (16 Aug), the Corsica bow echo (18 Aug), and the Bihucourt tornado (23 Oct).

Referee #2 also noted that there was a lack of clarity surrounding this decision. Thus, we will rephrase as follows :  "To target severe convective events, we restricted the selection to days where hailstones of size equal or greater to 2cm were reported in France in 2022, while keeping the number of cases reasonable in terms of computation and processing time. Thus, we included

all convective events from April to July, a period well characterized by severe convective storms and in particular supercells. In order to incorporate a variety of storm types, three high-stakes days were added to the study sample (a major thunderstorm event all across France on August 16, the Corsica bow echo on August 18, and the Bihucourt tornado on October 23), for a total of 34 convective days analyzed."

**L124:** I wonder whether AROME is perhaps predicting too many convective events in general because the events were selected for days on which thunderstorms actually occurred. It would be interesting to know whether AROME perhaps has a too high false alarm rate, i.e. whether too many thunderstorms were predicted even on calm days.

On days where there is no potential for strong thunderstorms, AROME does not predict high intensity storms. Similarly, on calm days, AROME does not forecast thunderstorms. In fact, the intensity dependence is quite robust. Overall, AROME has a frequency bias and predicts too many weak thunderstorms (in particular at night), but this does not apply to strong thunderstorms. As a result, it does not affect the choice of cases studied. This is illustrated by the enclosed plot.

[Figure]

⇦ Quantile-quantile diagram of AROME lightning density forecasts vs observed lightning density (from Météorage data), both evaluated on the same 10-km resolution grid, over June 2024.

SI = strike index (flashes per hour per square km)

Note : the figure is similar to other Spring months and years.

It shows that, on average, AROME predicts too many weak thunderstorms (QQ curve above the diagonal) where none is observed (a caveat being that some weak flashes may go undetected by the Météorage system), whereas AROME slightly underpredicts the frequency of the strongest thunderstorms (QQ curve under the diagonal for observed lightning densities larger than 0.8).

Since the paper is already quite long, we do not propose to include such a plot, but we propose to insert a sentence to summarize this information : "On average, AROME tends to slightly underpredict the frequency of the strongest thunderstorms (not shown)."

**L126:** So the newly calculated polarimetric data come from all three bands that were mixed together? Depending on the bands, the distance and the hydrometers present, the polarimetric signal could be different. What is selected then? I assume that ANTILOPE QPE takes this into account, but how is this taken into account in section 4.3?

Each polarimetric radar is processed through the polarimetric chain, our input radar data are hence already corrected accordingly to their frequencies, in particular for the attenuation correction (see Figueras i Ventura et al., 2012). However, we avoided using X-bands if they could have been included. So, for each event, we only used either exclusively C-bands or S-bands, or a mix of the two. We added a precision : "For the 3D evaluation, we only used C- and S-bands radar data, as X-band signal is quickly attenuated during rainy events."

Among the studied events (see Table A1 in the Appendix), S- and C-band may have been mixed for those that occurred in the southeast or south of France (where the S-band radars are located). In this case :

1.  during the interpolation step of the observations, contribution from each radar at a given grid point is weighted by its distance to the grid point. This means that, in the vicinity of an S-band radar, the resulting polarimetric field will be more influenced by the S-band radar than by a more distant C-band radar.
2.  when we applied the forward operator on model outputs, we have chosen the frequency band according to the majority radar band (so for example S band if there were a majority of S-band radars that contributed to the observational grid). This has been clarified at the end of section 3.1 : "Finally, the radar forward operator is applied to model outputs to simulate polarimetric fields, with the frequency band chosen in accordance to the majority radar band in the corresponding observational dataset."

However, we did not apply a correction factor to the output of the simulated S-band datasets to bring us back to the values of a C-band radar (or inversely). For example, as KDP is inversely proportional to the wavelength, a simple multiplication by two would allow us to homogenize KDP values of S-band ($\lambda \approx 10cm$) and C-band ($\lambda \approx 5cm$). But it is not that straightforward with ZH and ZDR. For subsequent work, we will keep in mind this idea, and apply correction if the number of S-band datasets is significant compared to C-band datasets.

**L255:** Is coverage by 3 different radar stations per grid box required? Or do you mean at least 3 radar observations per grid box? If 3 radar stations are really required, you would have to exclude a lot of data, or how dense is the radar network? And again, how are the S, C and X bands combined?

Here is an illustration of the French radar network, with radii of 100 km for C- and S-bands, and 50km for X-bands (for illustration purpose). The maximum range of our C- and S-bands radar is 255 km.

[Figure]

However, the higher the elevation angle, the lower the maximum range. To ensure enough coverage and avoid holes in the gridded data (especially on top at 10–13 km high), we determined that we needed at least the contribution of 3 radars on the considered domain (one domain per event), so that at least one elevation angle among all elevations available (from all radars) contributes to each grid box. This number was found to be a good compromise between data loss in the upper part of the grid and computational cost. Hence, we carefully selected the radars that were to contribute to the grid (in most cases we used more than 3). Since our events are centered on the domain, only the edges of the domain can suffer from gaps in the gridded data. We added "In other words, at least one elevation angle among all elevations available (from all radars) contributes to each grid point." in section 3.1 to make it clearer.

**L259:** Have you analyzed the impact of the lead time of your prediction? In other words, does it make a difference how long after the NWP start the event took place? I would imagine that your results could be different depending on the lead time, as the model tends to produce fuzzier predictions with longer lead times.

This is an excellent suggestion, as NWP forecasts are indeed usually evaluated in terms of their lead time (see for example Figure 10 of Brousseau et al., 2016 which show the HSS compared to persistence forecast averaged over different thresholds of 6h cumulative rainfall against rain-gauge measurements, as a function of the forecast range, for forecasts initialized at 0000 and 1200 UTC over a four-month convective period). However, we did not analyze the data given their lead time in this study, as the main objective was to focus on the ability of the model microphysics, combined with the forward operator, to accurately simulate the storm characteristics in terms of polarimetric radar data (and not in terms of location and temporality). Different configurations of AROME, with LIMA microphysics, initialized at 00 UTC are currently running in real time for a long-term evaluation over the whole France domain. With these new generated data, it will be possible to take lead time into account, but this is beyond the scope of the present study.

**L357:** 50 x 50 km box means that with the regrided radar resolution of 500m per lon/lat in each box there are 10000 values, right? And from that the 99 percentile is taken? And what is true for the model data with 1.3km resolution? Something like 1600 values per 50 x 50 km box? Why are you not using the same grid?

The horizontal resolution of our observation grid is 1 km (500m is on the vertical). So in a 50x50 km box, the 99th percentile is computed from 2500 obs values versus 1480 model values. However, what we are comparing in this section is AROME with ICE3 microphysics against AROME with LIMA microphysics. All contingency tables have been constructed identically, which means that the use of 1x1km observation QPE affects AROME-ICE3 and AROME-LIMA scores the same way.

**L365:** How is the bootstrapping done? Out of the 50x50 km boxes? How often?

We also calculated, for each rainfall threshold, the difference between the ICE3 and LIMA scores (i.e. $\Delta POD = POD_{LIMA} - POD_{ICE3}$, idem for FAR and HSS). The bootstrapping was performed on these $\Delta POD$/FAR/HSS values (for each threshold, one $\Delta$ value per case study = max 38 values). Note that for the highest rainfall thresholds, it was sometimes not possible to calculate a difference (e.g. no occurrences), which reduced the number of values used for the bootstrap (this number is shown in bold in our Figure 2). We used the `scipy.stats.bootstrap` function (see the description here) with `statistic=np.mean`, `confidence_level=0.95` and `method='basic'`. A precision has been added : "Additionally, a bootstrap test is performed using the scores' differences from all 38 cases to determine if the differences between ICE3 and LIMA are significant."

**L386/ Figure 3:** As far as I understand, there is a discrepancy between the resolution of the model and the observation grid. So does it make sense to compare the very small observed cells with the coarser model? I don't understand that here. And the discussion about the small cells being 'simulated too rarely' is then misleading. How should a model with a resolution of 1.3 km be able to simulate these very small cells (i.e. with a size of about 0 km^2)? I think there should be a discussion about model resolution in this section. The same goes for the discussion about lifetimes: I don't think AROME is able to predict the very small lifetimes. That is related to the model resolution.

We agree that our paper lack of a model resolution discussion. In our Figure 3, we displayed the convective cores size distribution, but you may be interested in the same plot for the cell envelope :

[Figure]

The difference in resolution between the observations and the model is unavoidable, as the best possible observations are usually preferred, especially in an assimilation context (which is our long-term goal). This paper is precisely about the possibility of an obs/model comparisons despite this inconsistency. However, Ricard et al. (2013) diagnosed the effective resolution of AROME to be of the order of 9-10Δx, where Δx=1.3km is the grid resolution. It implies that, in the above figure, objects of size smaller than 13x13 = 169km² cannot be fully resolved (neither in terms of size nor lifetime). A brief discussion will be added in section 4.2 : "This behavior is expected, as Ricard et al. (2013) diagnosed the effective resolution of AROME to be of the order of 9 − 10Δx, where Δx = 1.3 km is the grid resolution. This implies that objects of size smaller than 136 – 169 km2 cannot be fully resolved by AROME."

**L416:** Why should LIMA be compared to OBS_no_hail? LIMA is not directly including hail, but it is included in the graupel class.

This is right, in this study hail is not considered as a separate class but is instead included in the graupel class, which implies it has the parametrization of the graupel (same 1-moment PSD, fall speed, mass-size relationship, etc.).

Please find on the right an illustration of the theoretical output from the forward operator for ZH values as a function of the content (at C-band, for a 1-moment PSD and T=0°C).

Furthermore, the forward operator inherits the microphysics properties of the graupel to compute the resultant polarimetric values. How could we fairly compare observed reflectivities (due to rain and sometimes hail) with simulated reflectivity

values due to rain only ? Thus, we have chosen to distinguish between rain-only precipitation and total precipitation. We also added a precision regarding the "OBS_no_hail" distinction : "[Fig5] has been complemented by another curve (in gray) that only includes observed cores not associated with medium hail (5 − 20 mm) or large hail (> 20 mm) as detected by the A13 radar hydrometeor classification algorithm (see Appendix C of Forcadell et al., 2024)."

**L425/Figure 6:** Why CFADs instead of CFTDs (Contoured Frequency by Temperature Diagrams)? Maybe then the melting layer would be more sharp and the whole discussion in 4.3 more meaningful. I could imagine that the 44 events vary a lot in surface temperature and by that the distributions of observed pol. Moments in Fig. 6 are broader.

Thanks for this relevant question. We started the work with a grid in km altitude levels because of the radar data. The interpolation was easy from radar gates altitudes to regular gridded altitude data. Hence, we kept the altitude resolution for the model grid. As you rightly suggested, we are planning to switch to CFTDs for our current work on the forward operator, that more specifically focuses on stratiform events and bright band simulation.
* * *
**Technical corrections:**

L366/Figure 2: "The HSS score": leave out "score" as it is already in HSS. Done.

Figure 9: "Altitudes are given in km AGL": Is this a remark for Figure 8? Yes, now corrected.

---

## Author Comment (AC3)

**Review of egusphere-2025-685 #2**

**Indications of authors :** The authors would like to express their gratitude to the anonymous referee for his/her meticulous examination of the paper, which will undoubtedly enhance its quality. We hope that readers will find the answers useful.

The response to each comment will be displayed in blue. Citations to parts of the paper will be quoted, and, if necessary, modified text according to the reviewer's comments will be underlined.
* * *
**Summary**

This study presents an evaluation of the AROME model with two cloud microphysics schemes based on polarimetric radar observations over a dataset of 34 days of convective precipitation in France. The evaluation encompasses general precipitation metrics and convective cell characteristics, with a special focus on ZDR column statistics. Leveraging polarimetric radar observations for model evaluation is a promising approach, and this study benefits from a large sample size. It is remarkable that the model with LIMA is able to reproduce area and frequency statistics of ZDR columns that well as demonstrated by the authors. I find the paper to be well written, well-structured and with informative and well visualized images. The methods are explained thoroughly and clear. The content is extensive, resulting in a very long paper. I appreciate the discussion about limitations of the approach. In general, I think this paper is already of high quality, and my suggestions below are mainly to address clarity or for quality improvements.
* * *
**General comments**

1. At some instances, I find the result section to be rather descriptive and to sometimes lack possible explanations for the observed discrepancies. This concerns mainly section 4.2. It would be interesting to know about potential reasons for these discrepancies. This might be related to grid spacing or interpolation issues and should be discussed.
   Discussions about the interpolation or grid resolution effects were not the main topics of this paper. However, we added (when necessary) a few comments regarding these aspects.

2. I do not fully understand the focus on the problems regarding the bright band and melting scheme of the forward operator as it seems to me that this is not important for the purpose of this study. The uncertainties in that region are dominated by the forward operator and as such do not help much in evaluating the microphysics model. Perhaps you could add a paragraph explaining why you think this is important for your topic. Alternatively, less attention to this topic might be a way to reduce the length of this paper.
   We appreciate your suggestion. Reducing attention on the melting layer would render the 3D evaluation over the vertical less exhaustive, whereas we attempted to be as complete as possible.
* * *
**Specific comments**

1. **Line 6:** I think I disagree with the logic. I would argue it is the other way around: It is a complex issue, which results in a diversity of forward operators and microphysics schemes.

   It seems to me that both our visions illustrate the fact that, since no truly satisfactory solution has yet been found, there are various approaches in terms of microphysics and forward operator. I will invert the propositions in the sentence : "However, the modelling of polarimetric values and radar signatures such as the ZDR column (ZDRC) remains a complex issue, despite the diversity of microphysics schemes and forward operators, especially above the freezing level where too low values are often found."

2. **Lines 36 – 57:** This paragraph is well written, and I enjoyed reading it. However, I do not understand the focus on nowcasting in the scope of your paper. Generally, AROME is not used for nowcasting, is it? I do think there is plenty of motivation to analyze ZDR columns anyway, as is described, because of data assimilation, and because ZDR columns are an indicator to analyze the performance of models in terms of correct updraft characteristics.

   It exists a slightly different version of AROME that is used for nowcasting, called AROME-PI (see Auger et al., 2014). Although nowcasting is not the subject of this article, we thought it was important to highlight this other potential use of ZDR columns. I understand that the paragraph concerning nowcasting is slightly more extensive than the one discussing assimilation, and as such, it has been shortened.

3. **Line 118:** Why do you select only 34 of the 51 days with hailstones?

   We understand that this is confusing. In fact, it took a lot of time to get from the archives the radar data, and to replay them. Furthermore, we also had to reforecast each studied day two times (one per microphysics). Since we worked with a temporal resolution of five minutes (so our model data was consistent with our radar data), it generated a huge amount of data to analyze. Therefore, we decided to focus on events that occurred during spring and early summer (i.e. from April to July). However, 2022 had been a year rich in strong convection events, and because we were aiming to have a wide diversity of study cases, we finally decided to include 3 more study cases in our analysis sample : a major thunderstorm event all across France (16 Aug), the Corsica bow echo (18 Aug), and the Bihucourt tornado (23 Oct).

   Referee #1 also noted that there was a lack of clarity surrounding this decision. Thus, we will rephrase as follows : "To target severe convective events, we restricted the selection to days where hailstones of size equal or greater to 2cm were reported in France in 2022, while keeping the number of cases reasonable in terms of computation and processing time. Thus, we included all convective events from April to July, a period well characterized by severe convective storms and in particular supercells. In order to incorporate a variety of storm types, three high-stakes days were added to the study sample (a major thunderstorm event all across France on August 16, the Corsica bow echo on August 18, and the Bihucourt tornado on October 23), for a total of 34 convective days analyzed."

4. **Line 126:** How was the network in 2022, given your evaluation is done with data from 2022?

   The network was all the same, we will change the year to avoid confusion.

5. **Line 127:** Does the radar frequency have an effect on your evaluation? E.g., are ZDR columns or convective cells detected in X-band the same way as in S-band?

   This is a relevant question, however we forgot to mention that we avoided using X-bands if they could have been included. Referee #1 also pointed out a lack of information about how we combined the different radar frequencies. Here is our answer to referee #1 :

Each polarimetric radar is processed through the polarimetric chain, our input radar data are hence already corrected accordingly to their frequencies, in particular for the attenuation correction (see Figueras i Ventura et al., 2012). However, we avoided using X-bands if they could have been included. So, for each event, we only used either exclusively C-bands or S-bands, or a mix of the two. We added a precision : "For the 3D evaluation, we only used C- and S-bands radar data, as X-band signal is quickly attenuated during rainy events."

Among the studied events (see Table A1 in the Appendix), S- and C-band may have been mixed for those that occurred in the southeast or south of France (where the S-band radars are located). In this case :

1. during the interpolation step of the observations, contribution from each radar at a given grid point is weighted by its distance to the grid point. This means that, in the vicinity of an S-band radar, the resulting polarimetric field will be more influenced by the S-band radar than by a more distant C-band radar.

2. when we applied the forward operator on model outputs, we have chosen the frequency band according to the majority radar band (so for example S band if there were a majority of S-band radars that contributed to the observational grid). This has been clarified at the end of section 3.1 : "Finally, the radar forward operator is applied to model outputs to simulate polarimetric fields, with the frequency band chosen in accordance to the majority radar band in the corresponding observational dataset."

However, we did not apply a correction factor to the output of the simulated S-band datasets to bring us back to the values of a C-band radar (or inversely). For example, as KDP is inversely proportional to the wavelength, a simple multiplication by two would allow us to homogenize KDP values of S-band ($\lambda \approx 10$cm) and C-band ($\lambda \approx 5$cm). But it is not that straightforward with ZH and ZDR. For subsequent work, we will keep in mind this idea, and apply correction if the number of S-band datasets is significant compared to C-band datasets.

6. **Line 171 and 186:** What do you mean by flexible? I would suggest to either explain or omit this word.
By flexible we meant that each hydrometeor class in LIMA can be one or two moments. We will omit this word.

7. **Line 199:** If hail is included in the graupel species, what does that mean for the density of this particle class? Given that you evaluate hail events, I would expect differences to the observed events purely as a result of the particle property assumptions here. Is there a reason why you chose not to use hail as a sixth category?
The density of hail is here the one of the graupel class. For information, both PSDs follow a general gamma distribution, and the mass-diameter law is $m = aD^b$ where
- for graupel : $a = 19.6$ and $b = 2.8$
- for hail : $a = 470$ and $b = 3$.

Furthermore, we used hail events of the ESWD database only to objectively identify severe convection.
The one-moment microphysics (ICE3) that runs in the operational version of AROME does not include a specific hail class. To ensure a fair comparison, we compared ICE3 against a version of LIMA that has the same classes (as a bonus, there is a reduction in the computational cost). Initially, we had planned to add two more data sets, which would have been the ICE3 and LIMA versions with a separate hail class. Unfortunately, the forward operator has not yet been validated for hail, and we would have had twice as much data to process (34 days x 4 model datasets instead of 34 x 2), which would have further increased the production time for data and results, as well as the length of the paper. Such evaluations will be conducted in future studies. It is important to note that LIMA is intended to replace ICE3 in the future operational version of AROME, meaning

that the computational cost will be a key consideration. This is also why we chose to evaluate a partially 2-moment version of LIMA.

8. **Line 217:** Why is melting snow transferred to graupel instead of rain? Is this a common approach?
This is how the scheme was made, assuming that when snowflakes melts, liquid water progressively fill holes and surround the particle. In terms of density, it is similar to that of the melting graupel. In the case where this particle is brought back at colder altitudes, the melted snowflake will refreeze and may look more like a graupel particle. For both microphysics schemes (ICE3 and LIMA), snow cannot melt directly into rain but is first transferred into the graupel class (see Figure 7 of Lac et al., 2018). In contrast, for example, the actual microphysics running operationally in ICON allow melting of snowflakes into rain directly (see Table 2 of Seifert et al., 2006). To clarify this point, we added a sentence in section 2.3 : "A specificity of ICE3 and LIMA is that snow cannot be converted directly into rain when it melts, but instead is first converted into graupel."

9. **Line 249:** I do not understand how the nearest radar determines the ROI for a given grid point. How is the radar affecting the radius that is taken into account for interpolation to a given grid point? Especially since you use multiple radars for interpolation, but only one determines the ROI?
We also struggled to understand the reasons, but this is simply how Py-ART works. First, it creates an empty grid based on the user specifications (in our case a horizontal resolution of 1km and a vertical resolution of 500m). Then, it maps the location of the radars onto the grid. For each grid point, the ROI is estimated based on the closest radar and a built-in ROI function (see here). We tested the 3 built-in functions and decided to use the one where radius grows with the distance from each radar and which parameter are based on virtual beam sizes. Thus, a ROI value is associated to each grid point, describing a sphere centered on the grid point. The closer the radar gate is to the center of the sphere, the more weight it has in the interpolation process.
The referee is referred to Helmus and Collis (2016) for an extensive description of the interpolation performed by Py-ART. We added "Thus, at least one elevation angle among all elevations available (from all radars) contributes to each grid point." in section 3.1 for clarity.

10. **Line 256:** Do you mean three radars are required for each 3D grid point? Or only after projection to a 2D plane? I suspect the latter, because only then your argument with vertical coverage makes sense. Perhaps you can rephrase this to make it clearer.
We had to reprocess the raw polarimetric radar data for each event individually (radar by radar), as the processed data is not archived at a 5-minute temporal resolution. Sometimes, no data was available. To ensure enough coverage and avoid holes in the gridded data (especially on top at 10–13 km high), we determined that we needed at least the contribution of 3 radars on the considered domain (one domain per event), so at least one elevation angle contribute to each grid box. This number was found to be a good compromise between data loss at the upper part of the grid and computational cost. Hence, we carefully selected the radars that were to contribute to the grid (in most cases we used more than 3). Since our events are centered on the domain, only the edges of the domain can suffer from gaps in the gridded data. We added in section 3.1: "Thus, at least one elevation angle among all elevations available (from all radars) contributes to each grid point."
For information : a map of the French radar network is shown in referee #1 responses.

11. **Line 261:** The word 'constrained' sounds like you make the domain smaller. But I think your point is that you take an extra area around the observed event into account, to include also mislocated predicted convection. Perhaps rephrase this to make this more clear.

We rephrased this way : "the forecasts' outputs encompass the observed domain by ± 0.5 ° in latitude and longitude, and the observed duration by ± 2 hours. This approach was adopted in an attempt to include mislocated predicted convection."

12. **Line 264:** So the model output is at a different horizontal resolution than the radar observations. Also, given the much higher native resolution of the radar compared to the model, I would expect differences that are solely based on these resolution effects. This is never discussed in this paper, but might be important.

We agree that a discussion on the resolution effects is missing. The difference in resolution between the observations and the model is unavoidable, as the best possible observations are usually preferred, especially in an assimilation context (which is our long-term goal). This paper is precisely about the possibility of an obs/model comparisons despite this inconsistency. As it is already lengthy (and resolution is not the main topic of our paper) a few comments have been added in section 4.2. regarding

- the small convective cores : "This behavior is expected, as Ricard et al. (2013) diagnosed the effective resolution of AROME to be of the order of 9 − 10Δx, where Δx = 1.3 km is the grid resolution. This implies that objects of size smaller than 136 – 169 km2 cannot be fully resolved by AROME."
- the biggest convective structures : "Brousseau et al. (2016) also showed that AROME model tends to overestimate the objects' sizes over a sample of 48 convective days of 2012."

13. **Line 281:** Perhaps for clarification you could write 'top of the Zdr column'. This is clear from the context, but one might also think of the full model column here.
Done : "to the top of the ZDRC"

14. **Line 313:** What effect does that have? If you have a large precipitation area of > 36 dBZ, with multiple embedded cores of 40 dBZ < max reflectivity < 48 dBZ, then they would count multiple times for your simulations, but only one time for your observations. Am I understanding this right?
This is right. However, for the case you described, we mostly observed the inverse situation: multiple embedded cores were identified in the observations, but not within the simulations (cores were present but not intense enough to be detected). As a result, we would have to compare (for example) 7 identified cores in the observations versus 1 big identified core in the model (which may contain multiple embedded cores of maximum reflectivity lower than 48 dBZ that remain undetected).

If only the highest reflectivities are missing in simulations (e.g., due to density assumptions), then it would perhaps be fairer to lower the threshold for both, observations and simulations. Perhaps a simple histogram of the observed / simulated reflectivities could help identify which range of reflectivities is affected by this bias.

In my opinion, lower the threshold will allow the detection of very early or late stages of the convection, which will not add more occurrences of high ZH. Here is a distribution of the maximum reflectivity over the vertical for all study cases (all grid points, all times) :

[Figure]

Such a decreasing slope had also been observed by Brousseau et al. (2016) in their evaluation of AROME 1.3km (with ICE3) using the maximum reflectivity within basic cell objects (see their Figure

8, panel (e) ZH > 31 dBZ and (f) ZH > 41 dBZ). Here is the same figure with our data, with a selection of all maximum reflectivity grid points > 35 dBZ :

[Figure]

[Figure]

(left : linear y-axis  /  right : same but logarithmic y-axis)

With the left figure, we find a similar curve as Fig.8e of Brousseau et al. (2016) : the highest reflectivities are clearly underestimated in the model with ICE3, a little less with LIMA. Now in our paper : "The underestimation of the highest reflectivities in the model (in particular with ICE3 microphysics) had already been demonstrated by Brousseau et al. (2016) within 31 and 41 dBZ reflectivity contour objects (see their Figure 8, panel e and f)."

Perhaps you could also elaborate on why exactly you chose to reduce the threshold to 40 dBZ, and not any other number. And do you have an idea about the reason for the bias? I think this was not mentioned in the result section. I think this choice requires a bit more discussion about the reasoning and the implications.

In our study, the selection of thresholds is of the utmost importance because of the tracking step. Indeed, an untracked cell will not count into the statistics. The 40 dBZ threshold comes from a subjective analysis : for a few different cases (MCS, SC, unorganized convection, etc.) we compared the quality of the tracking performed by *tobac* (the tracking algorithm). We tested different parametrizations to finally chose the one that is summarized in Table C1 of Appendix C (displayed hereafter).

| | cell envelope | cell core | | $Z_{DR}$ column |
|---|---|---|---|---|
| | | *obs* | *model* | |
| **Features detection** | | | | |
| Threshold(s) | / | 36 and 48 dBZ | 36 and 40 dBZ | 500 m |
| Minimum object(s) size | / | 20 and 5 km$^2$ | 23 and 6.5 km$^2$ | 4 km$^2$ |
| **Segmentation** | | | | |
| Contour threshold | 25 dBZ | 40 dBZ | | 2 dB |
| **Tracking** | | | | |
| Search radius | / | 20 km | | 15 km |
| Memory | / | 5 min | | 20 min |
| Minimum lifetime | / | 45 min | | 5 min |

As this paper is quite lengthy, I decided not to illustrate the impacts of the thresholds on the tracking. Nevertheless, please find below a demonstration of the effects of the features' detection thresholds on centroid placement for a LIMA (upper row) and a ICE3 (bottom row) case study : black dots are the centroids previously detected, and the one associated with a number corresponds to the displayed time step (identically between the left and right column). Where would a forecaster place the centroids of these storms ? In my opinion, the left column gives satisfactory centroid placements, that's why the 40 dBZ threshold has been preferred instead of 45 or 48 dBZ.

[Figure]

15. **Line 319:** It is not clear to me what the objects are that you use for later analysis. Are these the features identified in the identification step with 36 and 40/48 dBZ? Or the areas as defined in the segmentation step with 40 dBZ?

These objects are technically the same. One subtlety of *tobac* (the tracking algorithm) is that it uses contiguous points of values greater than a given threshold to create a surface **only used to locate** the feature centroid. Associated to the feature, we have access to the number of contiguous points of the detected feature, which reflect the area of 48 dBZ contour in observations and 40 dBZ in simulations. But then, once each feature is identified, we could add a supplementary segmentation step, to recalculate the area (which is what we did for a fairer comparison). Hence, our core and envelope definitions are identical for both observations and model data.

In other words, one object = one feature = one number of contiguous points = one area. If we had to count a number of objects, we used the unique identification feature number, if we had to compare areas, then we used the area values of the corresponding object. We are aware that the relationship between thresholds and objects is not intuitive. To address this, we included a table in Appendix C that summarizes which thresholds have been used and at which step.

16. **Line 319:** Following up on the previous question, what happens if you identify a core with a maximum reflectivity of less than 40 dBZ? Then you have identified a core, but no area assigned to it, given that the threshold you use for segmentation is higher than 36 dBZ, in my understanding.

Absolutely. In this case, the cell is not considered to include a convective core, and is therefore not taken into account in the statistics.

17. **Line 329:** I assume you use a 2 dB threshold as described earlier for the segmentation process? This should be mentioned here.
   Not exactly. We perform the feature identification on our "ZDR column height" 2D field (in meter). This field comes from my own algorithm of ZDR column computation. In the paper it is written : "As for the cell tracking, a detection of the columns is applied on the ZDRC field (in meters) obtained from the algorithm described in subsection 3.2. All ZDRCs greater than 500 m are detected, and their respective areal extents are determined through the tobac segmentation process"

18. **Line 359:** I do not understand the reasoning here. Convective precipitation is indeed typically associated with intense precipitation. However, RRtot is the total accumulated precipitation over the entire event length, as you describe in line 352. That means RRtot is not directly an intensity measure, as an event of medium/low intensity that lasts over a long time period might also produce high RRtot values.
   It is true that RRtots are not a direct intensity measure, as high accumulation precipitations can be associated with different types of events (stationary moderate convection, stratiform rain, very intense fast-moving convection, etc.). Nevertheless, we selected days of strong convection only, and chose the 99th percentile metric to target the highest accumulation rates. Using boxes of 50x50km² and RRtot over a relatively long period (varying depending on the day) helped to evaluate forecasted cells despite uncertainties about their location and temporality in the model. We rephrased : "This percentile has been chosen (rather than e.g. the median) in order to focus the verification on the ability of the model to accurately forecast the largest total precipitation accumulations."

19. **Line 373:** This is a significant difference between model and observations of more than a factor of 10. Is there an explanation for this? Is this a known problem of the model?
   As you may have seen with Figure 3 and 4, our simulated thunderstorms are bigger and last longer than the observed ones, which could explain this difference.
   → See also Figure 7 of Brousseau et al. (2016) : (a) frequency bias, (c) probability of detection, and (e) false alarm rate as a function of the 0000 UTC forecast lead time for 6 h accumulated rainfalls greater than 10 mm
   - at the daily maximum of the 6 h convective precipitation (i.e. 1800UTC), AROME 1.3km overestimates the rainfalls by 16%
   - according to the author, this is a known bias : "The objective and subjective evaluations of the model after six years validated its added value in terms of timing, location and intensity of convective events compared to its coupling model, but also exhibited some weaknesses (e.g. positive bias of convective rain amounts; Stein, 2011)"

   To avoid ambiguity, we have moved this sentence to section 4.2 : "The total rainfall over all cases reached $1.40×10^5$ $m^3$ for LIMA and $1.29×10^5$ $m^3$ for ICE3, while only $0.95×10^4$ $m^3$ of rain has been observed (not shown). This is consistent with the overestimation of size and lifespan we observed for the detected cells. The difference between the observed and simulated amount of convective precipitation is a known positive bias of the AROME model (Stein, 2011)."

20. **Line 439:** Why is KDP increasing towards the ground in LIMA, even though the actual rain mass is decreasing? The explanation of KDP as a proxy for the amount of liquid water seems to hold only for ICE3, not for LIMA.
   KDP is known to be a proxy of the amount of liquid water. It is also related to the shape of the drops, more specifically the mean raindrop axis ratio (Jameson, 1985 ; see also Eq.7.17 of Bringi and Chandrasekar (2001)). It is not explicitly written in the paper, so I will add it for clarity : "More specifically, Jameson (1985) demonstrated that KDP can be linked to the product of liquid water content and the mass-weighted mean raindrop axis ratio, which explains the different behavior of LIMA in Fig. 6c. Indeed, [...]"

21. **Line 454:** Could there be other differences between observations and model purely as a result of interpolation issues? Perhaps this should be discussed, similar to grid resolution differences.
The radar gate polar resolution is 240m x 0.5°. To obtain a Cartesian grid :
   - data are smoothed with a median filter on 3 gates and 3 azimuths (as stated in section 3.1)
   - then interpolation is performed from polar to Cartesian in a 1km x 1km x 0.5km grid.
There is definitely a loss of information throughout the process, but we never tried to quantify it. Furthermore, it is impossible to have a perfect equivalence between observation and model data, especially on a 3D grid. It impacts maybe even more the bright band region, as this is a vertically localized phenomena. As a consequence, our choice of the grid vertical resolution for interpolation had introduced differences, but we never tried to quantify it. Nevertheless, as mentioned in our discussion section (5.1), our forward operator is currently tested in stratiform environments and compared to QVP radar data instead of regrided data.

22. **Part 4.3.2:** It seems to me that the melting scheme is producing unrealistically strong melting signals. Are the bright band signatures important for your study? Perhaps an analysis without the bright band and without a melting scheme might provide results just as good?
It is true that the melting scheme produces a strong signal, which is not necessarily expected for convective cases. However, if we suppress the melting scheme, we won't be able to represent the bright band either. In addition, we focused the present study on ZDR columns, but they would also disappear as a result. For this reason, in section 5.1 we discussed the possibility of distinguishing, within the forward operator, between melting snow and melting graupel (as a reminder, melting snow is included in the graupel class).

23. **Line 472:** How can spherical droplets produce a ZDR that is not 0?
This is a typo, it has been removed : "A weak positive ZDR value can be associated with dry snow or graupel (which can potentially start to melt), , and aggregated ice crystals"

24. **Line 468–477:** This is a forward operator issue. The simulated ZDR/KDP depends strongly on the orientation and shape of the particles. What does the forward operator assume here for graupel and snow?
The parametrization for dry graupel and dry snow can be found in Sections 3.3.2 and 3.3.3 of Augros et al. (2016) (AU16) but is hereafter summarized :
   - Dry snow
     - Axis ratio : a linear decrease from 1 to 0.75 for diameters from 0 to 8 mm and a constant axis ratio of 0.75 for diameters higher than 8 mm.
     - Oscillation : neglected.
     - Density diameter law : equivalent to the one from Locatelli and Hobbs (1974) for "aggregates of unrimed radiating assemblages of plates, side planes, bullets and columns" ($m = aD^b$ with $a = 0.02$ and $b = 1.9$).
   - Dry graupel
     - Axis ratio : a linear decrease from 1 to 0.85 for diameters from 0 to 10 mm and a constant axis ratio of 0.85 for diameters higher than 10 mm based on Ryzhkov et al. (2011) (RY11).
     - Oscillation : neglected as increasing the axis ratio from 0.75 (like in RY11) to 0.85 produced the same effect on ZDR and KDP values in AU16.
     - Density diameter law : $m = aD^b$ with $a = 19.6$ and $b = 2.8$.

This could also be related to the forward operator being based on the T-matrix, as the T-matrix assumes soft spheroids and hence cannot represent the density/shape of realistic aggregates which can strongly deviate from soft spheroids. You discuss this well in your discussion section 5.1. I would suggest to mention here that there is a discussion about this later on.

Thanks, this has been added : "A more exhaustive discussion of these aspects can be found in subsection 5.2."

25. **Line 484:** Is this really the main reason for the reflectivity differences? The ice class surely stands out in the mixing ratio plot (6d) / 6e), but mainly due to a high relative difference (basically no ice in ICE3). However, the simulated snow mixing ratio also deviates by a lot between the schemes. And at least the Q75 for graupel is significantly higher in LIMA too.

There is much more ice in ICE3 than in LIMA because (1) LIMA's ice production is limited by the availability of ice nuclei, and (2) the ice is converted to snow as soon as ice crystals grow larger than 125µm. This LIMA parametrization affect both the ice and snow contents. Not mentioning snow in the paper is a mistake, and it has been corrected : "the main difference between ICE3 and LIMA above the melting layer is the distribution of the ice and snow contents, whereas the median content of graupel is relatively similar."

Generally, the effect on reflectivity for a given mixing ratio depends a lot on the assumed density and the simulated particle sizes. To draw conclusions about reflectivity biases from mixing ratio differences without context about size distributions and density assumptions is difficult. Some radar forward operators distinguish between class contributions to the total signal in their output. Is it perhaps possible with radar forward operator applied here?

All PSDs follow a generalized gamma distribution and the mass-diameter relationship is given by $m = aD^b$.

- For ice crystals $a = 0.82$ and $b = 2.5$
- For rain $a = 524$ and $b = 3$
- For snow $a = 0.02$ and $b = 1.9$
- For graupel $a = 19.6$ and $b = 2.8$.

It is indeed possible to obtain the theoretical output values from the forward operator for each single class of hydrometeor. Please find hereafter ZH, ZDR and KDP as a function of the hydrometeor content for rain, snow, ice, and graupel class. In addition, the wet graupel content has been added for different liquid water fraction. This figure will be included in the Appendix to provide further illustration of the results.

[Figure]

26. **Line 514:** Do you have an idea about why cloud water is available with such high proportion within the identified ZDR columns? Cloud water should not increase Zdr, and as such not be relevant to ZDR column detection, in my understanding. Is there a physical reason why cloud water is enhanced within Zdr columns, e.g., updrafts?

Updrafts definitely increased the cloud water content locally in the ZDRCs. It is understood that cloud water should not increase the ZDR signal. However, the idea behind using cloud water in the

forward operator parametrization is to increase the liquid water fraction available for the wet graupel species, and thus the resultant ZDR values (as shown in the figure above).

27. **Line 524-531:** While I do agree that high ZDR is potentially a result of large raindrops, I am not sure about size-sorting as the reason, because the ZDR is already very high at 3.5–4 km. However, comparing the ZDR distribution to the mixing ratio of rain, it seems convincing that rain is generally responsible for the high ZDR values.

    This is simply a hypothesis. Given the fact that this in only observed within LIMA simulations (where rain is two-moment and hence able to produce bigger raindrops) it may be possible that the ZDR arc signature (where size-sorting occurs) overlap at low-levels with the ZDR column signature.

    I am also not sure about your argument regarding the melting of graupel. 1) ZDR values are already too large above the melting layer, and 2) if melting graupel is the reason for too large rain drops, also the ZDR values within the convective core (Figure 6b) would be too large too. Perhaps some updraft related process is producing rain drop sizes that are too large? I am not sure about the actual reason, but this would be interesting to discuss.

    This is a relevant observation. It is true we only observed this effect within ZDRC and not convective cores. Your explanation makes sense, and it is likely to be an updraft-related process. Indeed, convective core objects we detected also include some of the downdraft regions, where raindrops are no longer lifted by an updraft and subsequently fall. We don't have room to discuss this so we will just modify section 4.4 with "A likely explanation is that detected convective core objects also include some of the downdraft regions, where raindrops are no longer lifted by an updraft and subsequently fall, unlike updraft regions targeted by the ZDRC objects. Moreover, the size sorting usually occurs at low levels, whereas the overestimated ZDR values are here mainly located around 3 km high, further supporting the hypothesis of an updraft-related process."

28. **Line 549:** How did you come up with these numbers? Based on previous studies? Based on your own observations?

    This is purely based on my own observation regarding the data we had, and I am aware that this is an approximate vision.

29. **Line 561:** Was this discussed before? You say, 'to summarize', but I could not find a discussion about the lack of rain (relative to ICE3 and only in ZDRC, I assume) and the impact on liquid water fraction.

    This has been mentioned previously "simulated ZDR values (Fig. 8a) rapidly decrease above the freezing level as well as the rain content (dark blue curves in Fig. 8b and c)". Maybe "summarize" is misunderstanding, so I will rephrase with "conclude".

30. **Line 577:** Are you referring to the ZDRC? Within the convective cores, both schemes show the same rain mixing ratio distribution.

    ZDRC are indeed impacted by the low amount of rain around the freezing level, but the sentence lack of a precision. Now : "These issues are characterized by an overestimation of the bright band (BB) in the simulations within convective regions, as well as a notably low amount of rain at negative temperatures in both convective cores and ZDRC objects, which is particularly pronounced with LIMA microphysics."

31. **Line 668:** You applied the same threshold for all simulations/observations. That means a lower number of ZDRC in the ICE3 simulations is not purely a result of the threshold, but rather points to the inability of that model to produce high ZDR values in these columns.

    This is also true, but this paragraph is mainly about the ZDRC detection methodology. We have added to the following paragraph : "As the last point of this discussion, we highlighted the inability

of the AROME model, when combined with the ICE3 microphysics, to produce elevated ZDR values in the ZDRCs."

32. **Line 677:** If the detected ZDRCs consist mainly of wet graupel, this should be part of the ZDRC section (4.4). In that section, the reasons for the discrepancies should be discussed and the impact of wet graupel is not mentioned so far.

The word "composed" is maybe misleading here, as ZDR columns are detected thanks to high ZDR values, which themselves are due to our parametrization of wet graupel that artificially enhance the ZDR signal at the melting level. We replaced in line 677 : "the detected ZDRCs in the model are essentially  due to wet graupel"

33. **Line 677:** Following up, I would expect that graupel does not produce high ZDR. Why do you think that the detected ZDRCs consist of wet graupel? If that is the case, why is LIMA then detecting so much more ZDRC than ICE3, given that ICE3 has higher wet graupel water fraction? (Figure 8d)

This is right, dry graupel does not produce important ZDR signal. However, we are creating this artificial "wet graupel" class, by stealing the graupel content when some conditions are met (e.g. coexistence with rain). Because of that, we do have wet graupel inside the simulated ZDRC at such altitudes (which is not shown on Figure 8d,e because it is not a native content from the AROME model, but can be seen with the altitude of the liquid water fraction in Figure 8f). As answered before, thanks to a high wet graupel liquid water fraction, ZDR signal is enhanced in ICE3 at higher altitudes (> iso0°C) than LIMA, which also show that the detected updrafts in ICE3 are probably the strongest as they brought more rain water aloft. Then, there is a second effect that could hinder the detection of the ZDRC, this time in the warm phase (where enhanced ZDR values are essentially due to larger raindrops), because of the way I have designed the ZDRC calculation algorithm. Indeed, to be detected as a ZDRC, a column of enhanced ZDR must be greater than - or equal to - 2dB over the vertical (with a tolerance of 2 missing grid points, see more details in section 3.2 of our paper). But because of the small raindrops size and the evaporation in ICE3, from the freezing level to the ground, ZDR barely exceeds our 2dB detection threshold (contrary to LIMA), which means that some columns in ICE3 are undetected because of the overall low ZDR values in the warm phase.

34. **Line 687:** How do you know that the rain mixing ratios below freezing temperatures were insufficient? You don't have real measured rain mixing ratios. I agree that rain is probably not reaching up high enough given that the ZDR columns reach up higher in observations. But statements about biases in the actual mixing ratios are difficult to make in your context. If you think statements about actual mixing ratio biases are possible from your analysis, please elaborate.

The raindrop content in LIMA reached higher altitudes than in ICE3. However, this content is so small that it had no effect on the simulated ZDR values. Maybe the term "supercooled rain water" is misleading here.

35. **Line 709:** What could be reasons for this?

A likely explanation is that a kilometric-resolution model such as AROME, like most other convection-permitting NWP models, is only able to explicitly simulate convective structures larger than the grid resolution. Indeed, studies like Ricard et al. (2013) have diagnosed the effective resolution of AROME to be of the order of 9-10Δx, where Δx=1.3km is the grid resolution. This is linked to spatially smoothing effects of the model dynamics such as interpolators in the semi-Lagrangian advection. The inability of AROME to organize convection over the full range of spatial scales that occur in nature can explain why simulated convective cells tend to be larger than observed ones. This remains a complex known issue (e.g. Brousseau et al., 2016 : "[AROME 1.3km] underestimates the number of cells and their maximal reflectivity, and also overestimates their size."). Our results are only valid for our studied sample, they may not be easy to generalize to other types of weather events. Still, we added : Our results show that the model, regardless of

the microphysics, have difficulties simulating small and short-lived cells, which is a known issue due to the AROME effective resolution of 9 to 10Δx (Δx=1.3km). In contrast, larger storms are often overestimated."

36. **Figure 1:** The ZDR column contours are really hard to see. Perhaps a different color would help to distinguish them from the background map? I am not sure. Also, what are these the spherical grey dots with black contours? Generally, the best way to demonstrate the tracking behavior would be a video as a supplement. I am aware that this might be too much work, so this is just a suggestion.
This is a very good idea, we produced a GIF that will be added as a supplementary material and mentionned in the Figure 1 legend.

37. **Figure 5:** Reflectivities below 36 cannot exist, due to your tracking thresholds of at least 36 dBZ. Perhaps adjust the x-axis limits correspondingly, otherwise one might be confused by the fact that there are never reflectivities below 36 dBZ.
I think this is clear enough given the description of Figure 5 in the text ("A study of the $ZH_{max\_z}$ distribution within each cell core object is herein proposed, with Fig. 5"), but just in case, this has been precized right after : "Please note that reflectivities below 36 cannot exist, due to the tracking thresholds of at least 36 dBZ."

38. **Figure 5:** Is this the maximum cell reflectivity over the whole lifetime of the storm? Or does a cell contribute multiple times, like in Figure 3?
We modified the legend for clarity : "Time dimension is omitted. Results are given in terms of relative frequency : each bin value has been divided by the total number of values (see Figure 4 legend) for each data series, and multiplied by 100."

39. **Figure 5:** Perhaps the absolute frequency is more helpful than the relative frequency. I would expect the observations then to overestimate the smaller max reflectivity values. Is this the case?
We first looked at the absolute frequency, and there was no overestimation of the smaller maximum reflectivities. However, we found the relative frequency easier to interpret and to work with as there were two times less detected cores in the simulations than in the observations (46,598 detected cores in the observations vs 25,170 in ICE3 and 29,140 in LIMA simulations)

40. **Figure 5:** It is remarkable that small reflectivities are as frequent as high reflectivities, e.g., that this is more or less a Gaussian distribution. I would have expected the extremely high reflectivities to occur much more rarely than smaller max reflectivities. Is this a result of the events you chose? Or a result of the thresholds applied within the tracking?
The distribution is a direct consequence of building the sample as a set of severe convective thunderstorms with high reflectivities (as explained in section 2.1), and fulfilling multiple criteria for object selection (described in section 3.3). In a long-term climate of observations and model simulations, small reflectivities would indeed be much more frequent than high ones. Such statistics about severe thunderstorms would be overwhelmed by a majority of weaker precipitating events. As shown in a previous answer, a simple selection based on a threshold of maximum reflectivity gives a different histogram, where small reflectivities are more frequent than high reflectivities.

**Technical corrections**

1. Line 708: Our results show. Done
2. Figure 8: Description of d) missing. Done